# Cryo-EM structures of human zinc transporter ZnT7 reveal the mechanism of Zn²⁺ uptake into the Golgi apparatus

Han Ba Bui[1,2,12], Satoshi Watanabe [1,2,3,12], Norimichi Nomura [4], Kehong Liu[4], Tomoko Uemura[4], Michio Inoue[1], Akihisa Tsutsumi[5], Hiroyuki Fujita[6], Kengo Kinoshita[7,8], Yukinari Kato[9], So Iwata [4], Masahide Kikkawa [5] & Kenji Inaba [1,2,3,10,11] ✉

Zinc ions (Zn²⁺) are vital to most cells, with the intracellular concentrations of Zn²⁺ being tightly regulated by multiple zinc transporters located at the plasma and organelle membranes. We herein present the 2.2-3.1 Å-resolution cryo-EM structures of a Golgi-localized human Zn²⁺/H⁺ antiporter ZnT7 (hZnT7) in Zn²⁺-bound and unbound forms. Cryo-EM analyses show that hZnT7 exists as a dimer via tight interactions in both the cytosolic and transmembrane (TM) domains of two protomers, each of which contains a single Zn²⁺-binding site in its TM domain. hZnT7 undergoes a TM-helix rearrangement to create a negatively charged cytosolic cavity for Zn²⁺ entry in the inward-facing conformation and widens the luminal cavity for Zn²⁺ release in the outward-facing conformation. An exceptionally long cytosolic histidine-rich loop characteristic of hZnT7 binds two Zn²⁺ ions, seemingly facilitating Zn²⁺ recruitment to the TM metal transport pathway. These structures permit mechanisms of hZnT7-mediated Zn²⁺ uptake into the Golgi to be proposed.

Zinc ions (Zn²⁺) are an essential trace element that plays vital roles in the structure and function of various proteins, acting as either a cofactor essential for enzymatic reactions or as a structural element stabilizing protein folding. About 10% of the mammalian proteome is known to bind Zn²⁺[1]. Cellular zinc homeostasis involves the opposing actions of two families of zinc transporters, SLC30 (ZnTs) and SLC39 (ZIPs)[2]. The SLC30 family proteins transport Zn²⁺ from the cytosol to the extracellular space or intracellular compartments, thereby reducing cytosolic Zn²⁺ concentrations. In human cells, the SLC30 (ZnTs) family consists of 10 homologs, named ZnT1 to ZnT10, which primarily regulate the dynamics of intracellular and extracellular Zn²⁺[3,4]. One of these proteins, human ZnT7 (hZnT7), localizes in the Golgi membrane[5-8] and transports Zn²⁺ from the cytosol into the Golgi lumen for incorporation into newly synthesized zinc enzymes[3,4]. The presence of ZnT7 and other Golgi-resident ZnTs maintains the labile Zn²⁺ concentration in the Golgi at around 25 nM or higher[9]. ZnT7 functions as a Zn²⁺/H⁺ antiporter and uses proton motive force to transport Zn²⁺ from the cytosol to the Golgi lumen. Our latest study showed that

[1]Institute of Multidisciplinary Research for Advanced Materials, Tohoku University, Sendai, Miyagi 980-8577, Japan. [2]Department of Molecular and Chemical Life Sciences, Graduate School of Life Sciences, Tohoku University, Sendai, Miyagi 980-8577, Japan. [3]Department of Chemistry, Graduate School of Science, Tohoku University, Sendai, Miyagi 980-8578, Japan. [4]Department of Cell Biology, Graduate School of Medicine, Kyoto University, Kyoto 606-8501, Japan. [5]Graduate School of Medicine, The University of Tokyo, 7-3-1 Hongo, Bunkyo-ku, Tokyo 113-0033, Japan. [6]Advanced Research Laboratory, Canon Medical Systems Corporation, Otawara 324-8550, Japan. [7]Department of System Bioinformatics, Graduate School of Information Sciences, Tohoku University, Sendai, Miyagi 980-8579, Japan. [8]Department of Integrative Genomics, Tohoku Medical Megabank Organization, Tohoku University, Sendai, Miyagi 980-8573, Japan. [9]Graduate School of Medicine, Tohoku University, Sendai 980-8575, Japan. [10]Medical Institute of Bioregulation, Kyushu University, Fukuoka 812-8582, Japan. [11]Core Research for Evolutional Science and Technology (CREST), Japan Agency for Medical Research and Development (AMED), Chiyoda-ku, Tokyo, Japan. [12]These authors contributed equally: Han Ba Bui, Satoshi Watanabe. ✉e-mail: kenji.inaba.a1@tohoku.ac.jp

hZnT7 localizes at the proximal side of the Golgi and contributes to the regulation of the intracellular localization and traffic of ERp44, an ER-Golgi cycling chaperone[10].

Physiologically, hZnT7 plays essential roles in dietary zinc absorption and regulation of body adiposity[11]. Decrease of cellular $Zn^{2+}$ in the epithelium of the prostate was shown to be involved in the development of prostate cancer in mice, with apoptosis being prevented in TRAMP/$Znt7^{-/-}$ mice[12]. Therefore, a null mutation in the $Znt7$ gene accelerated prostate tumor formation in mice[12]. In addition, a deficiency in ZnT7 reduced lipid synthesis in adipocytes by inhibiting insulin-dependent Akt activation and glucose uptake[13].

Despite progress in studies of the physiology of ZnT7, its structure and $Zn^{2+}$ transport mechanism remain to be elucidated. Structures have been determined for several zinc transporters belonging to the Cation Diffusion Facilitator (CDF) superfamily, including *Escherichia coli* YiiP (EcYiiP)[14,15], *Shewanella oneidensis* YiiP (soYiiP)[16–18] and human ZnT8 (hZnT8)[19], all of which belong to the SLC30 family. Cryo-EM analysis of hZnT8, a $Zn^{2+}$/$H^+$ antiporter localized to the insulin secretory granules of pancreatic β cells, revealed its overall architecture and $Zn^{2+}$-binding sites[19]. Amino acid sequence alignment (Fig. 1) suggests significant differences between hZnT7 and hZnT8, especially in their overall architecture, length of their cytosolic histidine-rich loops (His-loop), and numbers of $Zn^{2+}$-binding sites. Because of its physiological

importance, hZnT7 is a particular target of structural and mechanistic studies.

This study presents high-resolution cryo-electron microscopy (cryo-EM) structures of hZnT7 in both its $Zn^{2+}$-bound and -unbound states. A monoclonal antibody Fab fragment that specifically and tightly binds to native-state hZnT7 was prepared, and the structures of the hZnT7-Fab complex were determined by single-particle cryo-EM analysis. Consequently, five different types of hZnT7 conformations were illuminated; $Zn^{2+}$-unbound homodimer with both protomers in outward-facing (OF) form, $Zn^{2+}$-unbound heterodimer comprising protomers in OF and inward-facing (IF) forms individually, and three types of $Zn^{2+}$-bound dimers with different $Zn^{2+}$-coordination structures and transmembrane (TM) helix arrangements. Determinations of these distinct structures provided essential insight into mechanisms of hZnT7-mediated $Zn^{2+}$ transport into the Golgi lumen. Notably, His164, a His-residue within the His-loop, transiently engages in $Zn^{2+}$ coordination at the TM $Zn^{2+}$-binding site, followed by the replacement with His240, one of the His-residues that constitute a zinc-binding HDHD motif in the TM domain, suggesting the role of the His-loop for efficient $Zn^{2+}$ recruitment to the $Zn^{2+}$ transport pathway. The present findings reveal the structural and mechanistic features of hZnT7 in comparison with those of other $Zn^{2+}$ transporters of known structure, and have physiological implications for $Zn^{2+}$ homeostasis in the Golgi apparatus.

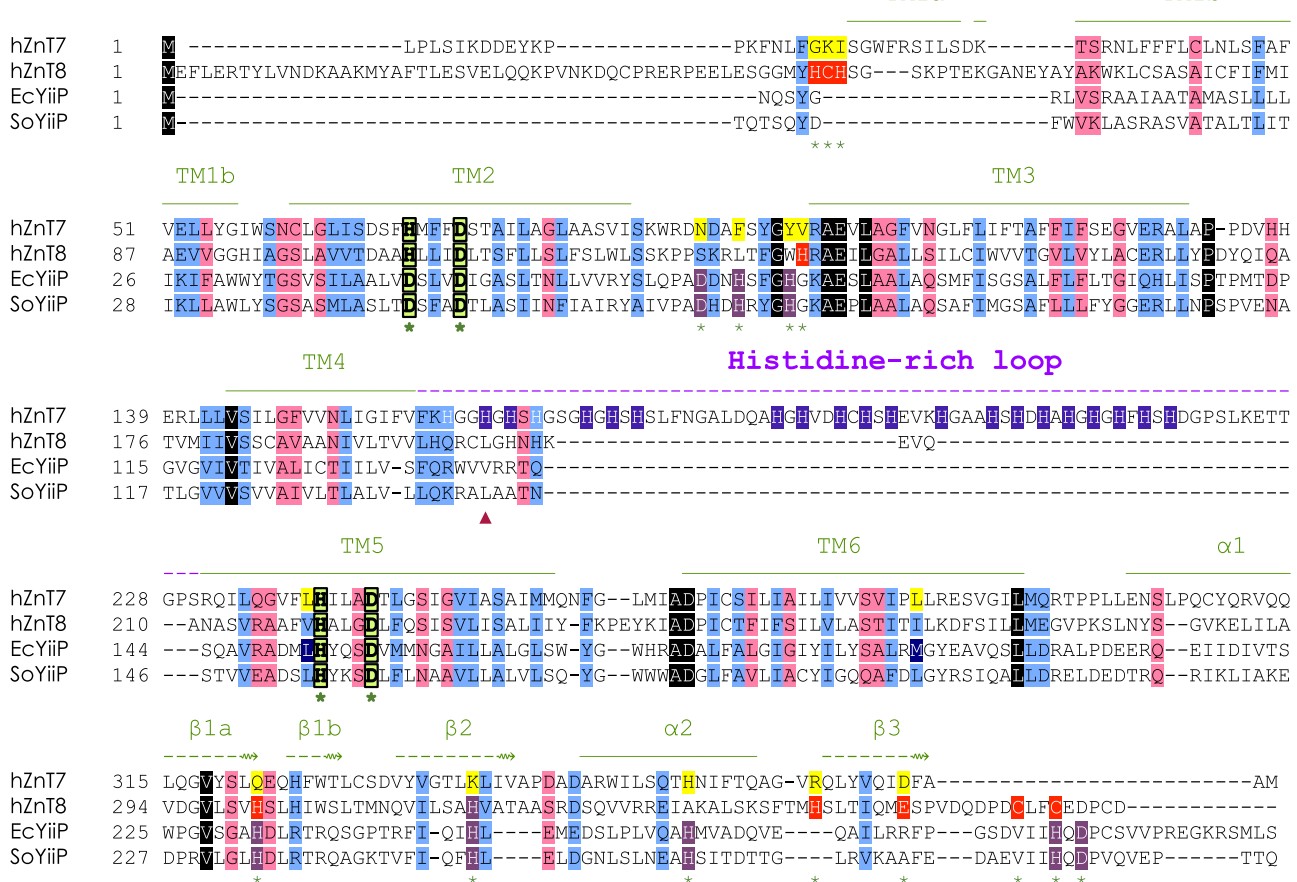

**Fig. 1 | Sequence alignment of representative ZnT-family members of known structure.** Amino acid sequences of human ZnT7 (hZnT7, UniProt ID: Q8NEW0) and human ZnT8 (hZnT8, UniProt ID: Q8IWU4), *Escherichia coli* YiiP (EcYiiP, UniProt ID: P69380) and *Shewanella oneidensis* YiiP (SoYiiP, UniProt ID: Q8E919) were aligned. Highly conserved, similar, and weakly conserved residues are colored black, light blue, and rose, respectively. The α-helix and β-strand elements of hZnT7 are indicated by solid and broken green lines above the amino acid sequence, respectively. The histidine-rich loop (His-loop) is indicated by a broken purple line above the sequence. HDHD-motif residues present in TM2 and TM5 are indicated by light green boxes. $Zn^{2+}$-binding residues present in the cytosolic domain of hZnT8 and Ec/SoYiiP are indicated by red and violet boxes, respectively; and histidine residues in the His-loop of ZnT7 are indicated by blue boxes. Green stars indicate residues involved in $Zn^{2+}$ binding. The red triangle indicates His164. TM, transmembrane helix; α and β, alpha-helix and beta-strand in the cytosolic domain, respectively.

## Results

### Structure of Zn²⁺-unbound hZnT7 homodimer

We employed single-particle cryo-electron microscopy (cryo-EM) analysis to determine the structure of the 74-kDa hZnT7 homodimer. After data processing of the repeated 2D- and 3D classifications, a density map of the hZnT7 homodimer was obtained at ~12 Å resolution, which did not allow de novo model building. This problem was overcome by preparing monoclonal antibody Fab fragments against recombinant hZnT7 to increase the molecular size and thereby generate higher-resolution cryo-EM maps (see "Methods" section). After thorough screening by ELISA and size-exclusion chromatography (SEC), four candidate monoclonal antibodies were selected, their Fab fragments (Fab#1, #3, #4 and #5) were generated, and four types of hZnT7-Fab complexes were prepared (Supplementary Fig. 1a-c). Two of these complexes, hZnT7-Fab#1 and hZnT7-Fab#3, yielded high-quality two-dimensional class averages by negative-stain EM measurements, showing that a Fab fragment symmetrically bound to each subunit of the hZnT7 homodimer (Supplementary Fig. 1d). The hZnT7 proteins in complex with Fab#1 and Fab#3 were analyzed by Talos Arctica Cryo-Transmission Electron Microscopy (Cryo-TEM), yielding 5.0 Å- and 7.7 Å-resolution density maps, respectively (Supplementary Fig. 2a, b). These results indicated that Fab#1 was the optimal binder of hZnT7 for cryo-EM analysis. The hZnT7-Fab#1 complex was therefore assessed by high-end Cryo-TEM with a Gatan K3 BioQuantum direct electron detector.

To prepare Zn²⁺-unbound hZnT7 uniformly, we purified hZnT7 throughout in the presence of 1 mM EDTA, and then assessed the structure of the hZnT7-Fab#1 complex using the JEOL CRYO ARM 300II. As a result, we determined the cryo-EM structure of the complex with a nominal resolution of 2.2 Å (Fig. 2a, Supplementary Fig. 3a-c, e left and Supplementary Table 1), which showed clear side-chain density for both the transmembrane domain (TMD) and cytosolic domain (CTD) (Supplementary Fig. 4a). Thus, hZnT7 was found to adopt a "*mushroom*"-shaped dimeric architecture with the TMDs and CTDs of two protomers tightly packed (Fig. 2c), which is distinguishable from the "*V*"-shaped dimers of hZnT8 and EcYiiP with less tight interactions between the TMDs of two protomers (Supplementary Fig. 5)[14,19]. On the other hand, SoYiiP forms a dimer via tight inter-protomer interactions[16–18], similar to hZnT7 (Supplementary Fig. 5). The TMD of each hZnT7 protomer contains six transmembrane helices (TM1b-TM6), with both the amino and carboxyl termini exposed to the cytosol (Fig. 2b, c) and forms a Zn²⁺ transport pathway like that of the hZnT8 and bacterial YiiP protomer[14,19]. TM helices are tightly bundled at the cytosolic half of TMD in both protomers, whereas they are widely opened to encompass a deep cavity at the Golgi luminal half, forming a solvent-accessible passageway from the luminal side (Fig. 2c-d). These structural features are typical of the outward-facing (OF) form of membrane transporters. Thus, the major portion of Zn²⁺-unbound hZnT7 has been found to form a homodimer composed of two OF protomers (named OF-OF homodimer). Structural alignments of hZnT7 with hZnT8 and YiiP show that TM1 and TM2 of hZnT7 move away from the rest of TMs to greater extent than those of hZnT8 and YiiP, forming the wider luminal cavity in the OF protomer (Supplementary Fig. 6a).

At the dimer interface of hZnT7, TM2 of one protomer tightly interacts with TM3 of another protomer near the luminal side (Fig. 2c and Supplementary Fig. 7a). In line with this, TM2 and TM3 contain numerous aromatic residues that interdigitate with each other at the dimer interface (Supplementary Fig. 7a). This mode of interaction is not seen at the dimer interface of hZnT8 or EcYiiP, likely leading to the less tight interactions between their TMDs (Supplementary Fig. 5 and 8b, c). Being unique to hZnT7, a membrane-parallel short α-helix (TM1a) is juxtaposed to the N-terminus of TM1b (Fig. 2b, c and Supplementary Fig. 8a), potentially contributing to maintaining or controlling the TM-helix arrangement from the outside. Additionally, the

present cryo-EM analysis revealed that the His-loop flanked by TM4 and TM5 is located on the cytosolic side (Fig. 2a, b). While the major portion of the His-loop was invisible probably due to its highly flexible nature, significant density was seen near the C-terminal end of TM4 (residues 159-163) and the N-terminal end of TM5 (residues 229-232), enabling the model building for the corresponding His-loop segments (Fig. 2e).

The cytosolic domain (CTD) of hZnT7 consists of residues 302-376 and follows the C-terminus of TM6 via a short linker composed of residues Met294 to Leu301. Cryo-EM analysis revealed that the CTD comprises two α helices (α1 and α2) and four β-strands (β1a, β1b, β2 and β3) (Figs. 1 and 2b), with the latter providing a contact surface for dimerization (Fig. 2f). An EQ linker (Glu323 and Gln324) flanked by β1a and β1b is conserved among hZnT7 orthologues and participates in a hydrogen bond network, along with neighboring residues of another protomer (Fig. 2f). The main chain of Glu323 and the side chain of His325 form hydrogen bonds to Tyr368 of another protomer. Tyr98 is hydrogen bonded to Gln366 (α2-β3 loop) and forms a π-π stack with Trp327 (β1b) at the TMD-CTD interface. Additionally, TM3 interacts with TM6 near the cytosolic end as also seen in hZnT8 and bacterial YiiP (EcYiiP and SoYiiP) (Supplementary Fig. 8a-d), via polar interactions between Arg102 (TM3) and Gln295 (TM6-CTD linker) (Supplementary 7c and 8d). The Fab fragment is specifically bound to the α1-β1a and β2-α2 loops at the end of the hZnT7 CTD (Supplementary Fig. 9). No direct interactions are made between the Fab fragment and the TMD or the TMD-CTD intermediate regions of hZnT7. It seems unlikely that Fab binding considerably affects the overall fold and TM helix arrangement of hZnT7.

### Structure of Zn²⁺-unbound hZnT7 heterodimer

It is notable that 3D variability analysis yielded a minor hZnT7 heterodimer composed of inward-facing (IF) and OF protomers at 2.8 Å resolution (Fig. 3a, b, Supplementary Fig. 3a, d, e right and Supplementary Table 1). The density map showed no Zn²⁺ bound to either protomer. Thus, some portion of Zn²⁺-unbound hZnT7 has been found to form a heterodimer composed of IF and OF protomers (named IF-OF heterodimer). Notably, a similar heterogeneous conformation was also observed in the structure of hZnT8[19]. TMs 1-6 of the OF protomer in the heterogeneous dimer are virtually superimposable with those of the OF protomer in the homogenous dimer (Supplementary Fig. 10a), suggesting that the structure of one protomer is hardly affected by the conformation of another protomer. In the IF protomer, TMs 2-6 are tightly bundled at the luminal half of the TMD, blocking the passage of solvent from the Golgi luminal side (Fig. 3c), whereas TM1b and TM4 are located more separately from the rest of TMs to create the wider cytosolic cavity in the IF protomer (Fig. 3d), while the OF protomer create the wider Golgi luminal cavity (Fig. 3e). During the transition from the OF to IF conformations, indeed, TM1 and TM2 considerably incline, and TM4 undergoes parallel shift so that the Zn²⁺ transport pathway opens at the cytosolic side concomitant with closing at the luminal side (Fig. 3f). By contrast, TM3, TM6, and the CTD remain almost static during this transition (Fig. 3f, Supplementary Fig. 11 and Supplementary Movie 1).

We superimposed the TMD of hZnT7, hZnT8 and SoYiiP in their IF conformations and found marginal differences between them, except for the orientations of TM1 and TM5 (Supplementary Fig. 6b). In hZnT7, TM1 swings away from TM2 while TM5 is located closer to the center of the TMD on the cytosolic side, compared to those of hZnT8 (Supplementary Fig. 6b left). Similar observations were made in comparison between hZnT7 and SoYiiP (Supplementary Fig. 6b right). Thus, hZnT7 appears to exert a different mechanism of action to open the cytosolic gate in the IF conformation than hZnT8 and SoYiiP, as further described in the next section. In line with this, we observed the different movements of TM4 and TM5 between hZnT7, hZnT8 and bacterial YiiP during the conversion between the OF and IF

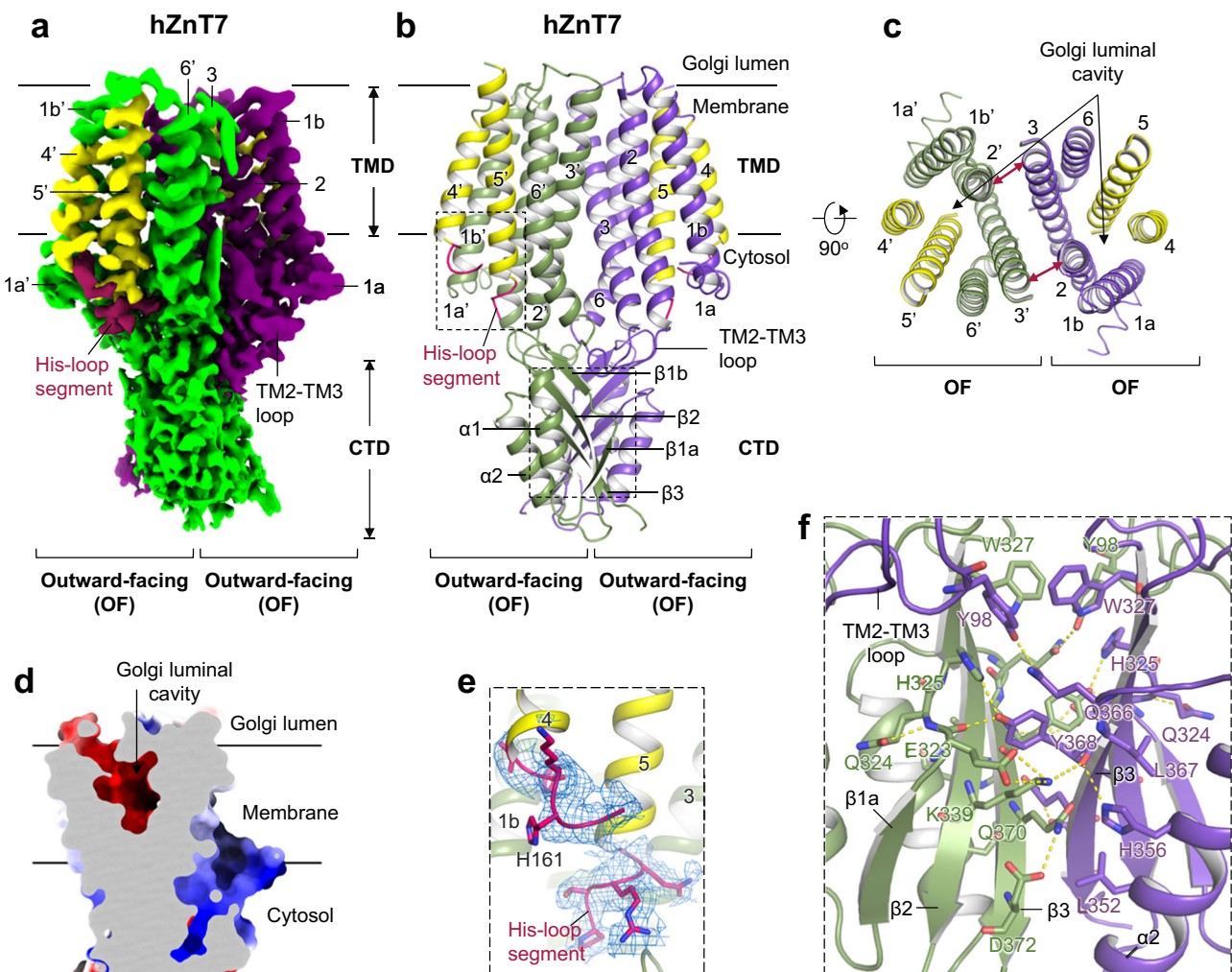

**Fig. 2 | Cryo-EM structure of the Zn²⁺-unbound hZnT7 homodimer composed of the outward-facing (OF) protomers (OF-OF form). a** Side view of the cryo-EM map of Zn²⁺-unbound hZnT7 homodimer. Chain A (violet), chain B (green), TMs 4-5 (yellow) and His-loop segment (magenta) are highlighted. Numbers indicate the transmembrane (TM) helix number from the N-terminus. TMD, transmembrane domain; CTD, cytosolic domain. **b** Cartoon representation of the hZnT7 homodimer in the same orientation as in (**a**). Both left and right protomers adopt an OF conformation. TM1 to TM3 and TM6 are shown in green or violet, respectively, while TM4 and TM5 are both in yellow. Visible regions of the histidine-rich loop (His-loop) are shown in magenta. Numbers indicate the TM helix number from the N-terminus. The black dashed-line squares indicate the regions that are highlighted in panel (**f**). TMD, transmembrane domain; CTD, cytosolic domain. **c** Golgi luminal

view of the TMD of hZnT7, where the CTD and TM loops are removed for clarity. The red double-headed arrows indicate the proximity between TM2 and TM3' (or TM2' and TM3) at the dimer interface. Numbers indicate the TM helix number from the N-terminus. **d** Coronal section of the electrostatic potential surface map of an hZnT7 protomer in the OF form. Surface colors indicate Coulombic potentials (red, negative; white, neutral; blue, positive). **e** Zoomed-in view of the visible segments of the His-loop near the C-terminus of TM4 and the N-terminus of TM5. TM4 and TM5 are represented by yellow ribbon. The density map is shown by blue mesh at a contour level of 0.17σ. **f** Zoomed-in view of the CTD-CTD interface in the OF-OF homodimer of hZnT7. Yellow dotted lines indicate interactions formed between polar residues.

conformations. As aforementioned, TM4 of hZnT7 shifts outward via parallel movement, whereas its TM5 moves only slightly (Fig. 3f). In hZnT8 and bacterial YiiP, both TM4 and TM5 swing using their luminal ends as pivot points, resulting in a wider cytosolic cavity in the IF conformation[14,19] (Supplementary Fig. 12).

**Structures of Zn²⁺-bound hZnT7 dimers**
We next determined the cryo-EM structure of Zn²⁺-bound hZnT7 in complex with Fab#1. To this end, we prepared Zn²⁺-bound hZnT7 in two different ways and found that these two preparations generated different dimeric structures of hZnT7. Concretely, the sample purified throughout in buffer containing 10 μM Zn²⁺ yielded a 2.7 Å-resolution cryo-EM structure of a heterogeneous hZnT7 dimer composed of Zn²⁺-bound IF and Zn²⁺-unbound OF protomers (named IF/Zn-OF heterodimer or Zn²⁺ state 1) (Fig. 4a left, Supplementary Fig. 13 and 14a left and Supplementary Table 1). Another hZnT7 preparation, which was

sampled by exogenously adding 200 μM or 300 μM Zn²⁺ to the above hZnT7 preparation, generated two types of dimeric structures: 2.9 Å-resolution cryo-EM structure of a heterogenous dimer with one protomer in Zn²⁺-bound IF conformation and the other in Zn²⁺-bound OF conformation (named IF/Zn-OF/Zn heterodimer or Zn²⁺ state 2) (Fig. 4a middle, Supplementary Fig. 14a middle and 15a, c, e and Supplementary Table 1), and 3.1 Å-resolution one of a homogenous dimer with both protomers in Zn²⁺-bound OF conformation (named OF/Zn-OF/Zn homodimer or Zn²⁺ state 3) (Fig. 4a right, Supplementary Fig. 14a right and 15b, d, f and Supplementary Table 1). The cross sections of the electrostatic potential surface demonstrate that the IF protomers of the Zn²⁺ states 1 and 2 have an even wider and deeper cytosolic cavity than the Zn²⁺-unbound IF protomer (Fig. 4b), whereas the OF protomer of the Zn²⁺ state 3 possesses a closed cytosolic cavity (Fig. 4b). Notably, the cytosolic cavity of the Zn²⁺ state 1 is strongly negatively charged with the open cytosolic cavity (Fig. 4c) compared with that of the Zn²⁺-

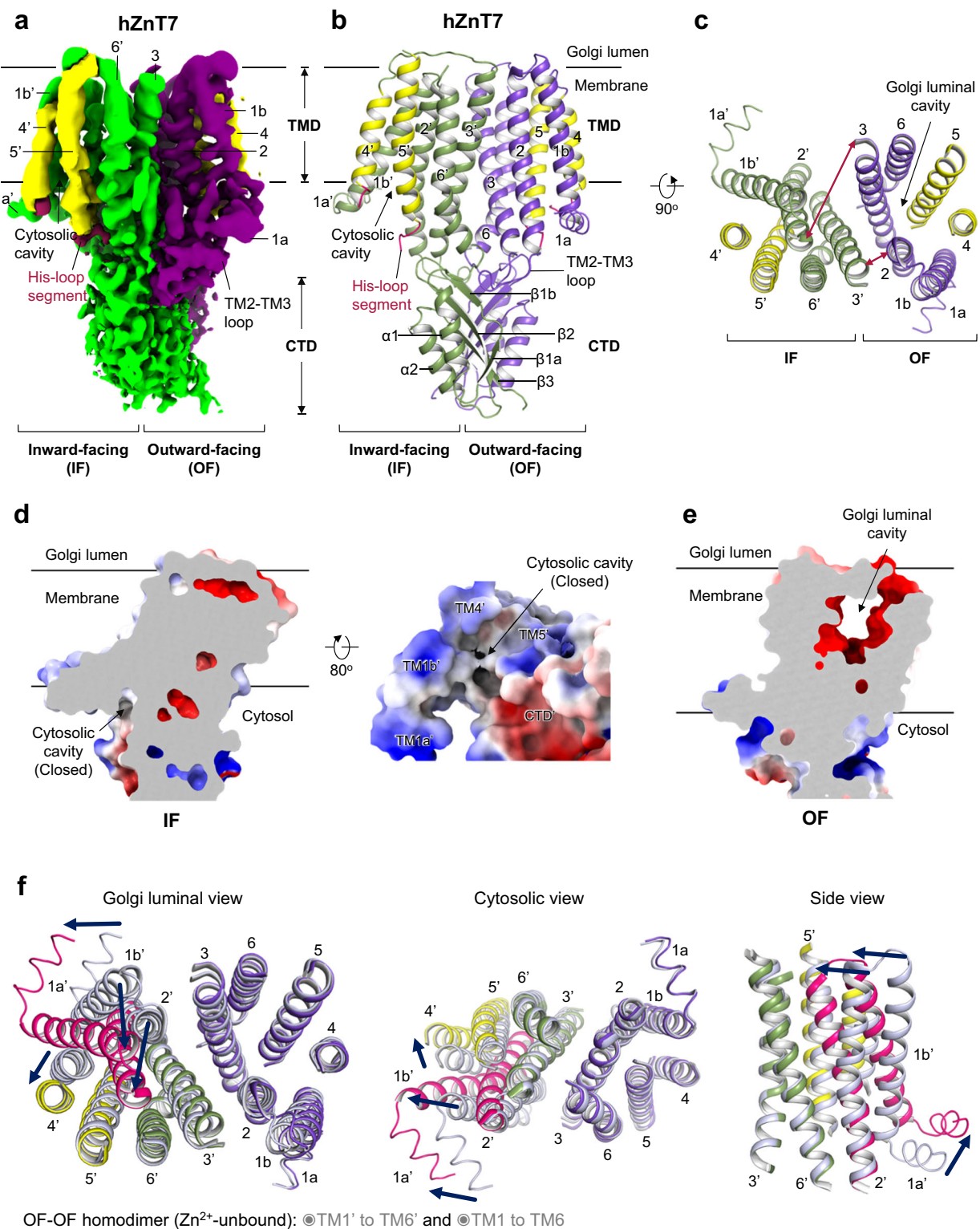

**f**

OF-OF homodimer (Zn²⁺-unbound): ◉TM1' to TM6' and ◉TM1 to TM6
IF-OF heterodimer (Zn²⁺-unbound): ◉TM1'-TM2'; ◉TM3'-TM6'; ◉TM4'-TM5' (in IF) and ◉TM1 to TM6 (in OF)

unbound IF protomer (Fig. 3d), which seems key to efficient recruitment of Zn²⁺ to the metal transport pathway in the TMD.

Superimposition of the Zn²⁺ states 1 and 2 reveals that there is no large conformational difference in the TMD between these two (Supplementary Fig. 10b). Similarly, the Zn²⁺-unbound heterodimer also shows good structural alignment with the Zn²⁺ states 1 and 2 (Supplementary Fig. 10c). The TMD of the Zn²⁺-unbound homodimer also

displayed a highly similar TM-helix arrangement to that of Zn²⁺-bound homodimer (Supplementary Fig. 10d). However, the IF protomer of the Zn²⁺ state 1 has a notable structural feature in the TMD; TM5 is kinked at Thr245 by 67° compared to that of the Zn²⁺-unbound IF protomer (Fig. 5a-c and Supplementary Movie 2). As a result, His240 is moved away from the Zn²⁺-binding site in the TMD (Fig. 5b, c). Instead, His164 contained in the His-loop is located nearby Zn²⁺ to be coordinated to

**Fig. 3 | Cryo-EM structure of the Zn$^{2+}$-unbound hZnT7 heterodimer composed of the inward-facing (IF) and OF protomers (IF-OF form). a** Side view of the cryo-EM map of Zn$^{2+}$-unbound hZnT7 heterodimer. Chain A (violet), chain B (green), TMs 4-5 (yellow) and histidine-rich segment (magenta) are highlighted. Numbers indicate the transmembrane (TM) helix number from the N-terminus. TMD, transmembrane domain; CTD, cytosolic domain. **b** Cartoon representation of the hZnT7 heterodimer in the same orientation as in (**a**). Left and right protomers adopt IF and OF conformations, respectively. TM1 to TM3 and TM6 are shown in green or violet, respectively, while TM4 and TM5 are both in yellow. Visible regions of the histidine-rich loop (His-loop) are shown in magenta. Numbers indicate the TM helix number from the N-terminus. TMD, transmembrane domain; CTD, cytosolic domain. **c** Golgi luminal view of the TMD of hZnT7, where the CTD and TM loops are

removed for clarity. The red double-headed arrows indicate the proximity between TM2 and TM3' (or TM2' and TM3) at the dimer interface. Numbers indicate the TM helix number from the N-terminus. **d** Coronal section of the electrostatic potential map of the IF protomer in the Zn$^{2+}$-unbound hZnT7 heterodimer. Surface colors indicate Coulombic potentials (red, negative; white, neutral; blue, positive). **e** Coronal section of the electrostatic potential map of the OF protomer in the Zn$^{2+}$-unbound hZnT7 heterodimer. **f** Superimposition of the Zn$^{2+}$-unbound OF-OF homodimer (gray) and IF-OF heterodimer (colored as indicated) of hZnT7 viewed from the Golgi lumen (left), cytosol (middle), and side (right). The cytosolic domain and TM loops are omitted for clarity. Numbers indicate the TM helix number from the N-terminus. Dark blue arrows indicate the movements of TM helices during the conversion from the OF to IF conformations in the left protomer.

the metal ion (Fig. 5e). In this connection, a His-loop segment consisting of residues Val158 to Gly165 is inserted into the cytosolic cavity (Fig. 5e). Of further note, the kinked segment of TM5 gets disordered in the IF protomer of the Zn$^{2+}$ state 2, and His240 moves back to be coordinated to Zn$^{2+}$ in place of His164 (Fig. 5d). These structural findings suggest a unique mechanism of the Zn$^{2+}$ uptake by hZnT7; His164 located near the N-terminus of the His-loop serves to recruit Zn$^{2+}$ to the negatively charged cytosolic cavity via the bending of TM5, followed by the exchange of a Zn$^{2+}$-coordinating residue from His164 to His240 at the TM Zn$^{2+}$-binding site (for more details, see the next section and Discussion).

### Zinc-coordination structures in hZnT7

The cryo-EM map of Zn$^{2+}$-bound hZnT7 demonstrated that the primary Zn$^{2+}$-binding site (hereafter, named S$_{TM}$) is located at around the halfway point of the TMD (Fig. 4a). At the S$_{TM}$ in the IF protomer of the Zn$^{2+}$ state 1, a tetrahedral Zn$^{2+}$ complex is formed by His70 (TM2) and Asp74 (TM2), Asp244 (TM5), and His164 (His-loop) while His240 (TM5) moves away from the S$_{TM}$ due to the bending of the N-terminal segment of TM5 (Fig. 6a and Supplementary Fig. 16a), as aforementioned. In the IF protomer of Zn$^{2+}$ state 2, His164 is replaced with His240 to form a similar tetrahedral complex to that in the Zn$^{2+}$ state 1 (Fig. 6b and Supplementary Fig. 16b). In the OF protomer of the Zn$^{2+}$ state 2, the side chain of His70 is moved away from Zn$^{2+}$ (Fig. 6d and Supplementary Fig. 16d), concomitant with significant increase of the distance between the C$_\alpha$ atoms of His70 and Asp244, seemingly facilitating the release of Zn$^{2+}$ to the Golgi luminal side. In support of this, His70 appears to further swing away from the S$_{TM}$ in the Zn$^{2+}$-unbound OF protomer with formation of a hydrogen bond between Asp74 and His240 (Fig. 6e and Supplementary Fig. 16e). In the OF protomer of the Zn$^{2+}$ state 3, the N$_\varepsilon$ atom of His70 is located beyond the range of direct Zn$^{2+}$ coordination (Fig. 6c and Supplementary Fig. 16c), and some extra density is seen between His70 and Zn$^{2+}$, suggesting the presence of a water molecule mediating the interaction between these two. Thus, the sidechain of His70 appears to be well separated from Zn$^{2+}$ and mobile in the Zn$^{2+}$-bound OF protomer, possibly representing an initial stage in rehydration of the metal ion prior to its release.

The previously published structures of the Zn$^{2+}$-bound OF form of EcYiiP, the Zn$^{2+}$-bound IF form of SoYiiP, and the Zn$^{2+}$-bound OF form of hZnT8 demonstrate the presence of additional Zn$^{2+}$-binding sites (Supplementary Fig. 17a). A second Zn$^{2+}$-binding site of hZnT8 is located at the TMD-CTD interface (S$_{IF}$) (Supplementary Fig. 17b), although this site had a lower affinity for Zn$^{2+}$[19]. Similarly, Zn$^{2+}$-binding sites are present near the TMD-CTD interface of EcYiiP and SoYiiP (TM2-TM3 loop). hZnT7 lacks the second Zn$^{2+}$-binding site around the TMD-CTD interface, consistent with the absence of histidine residues in this region (Supplementary Fig. 17b).

In contrast to hZnT7, hZnT8 and bacterial YiiP have two additional Zn$^{2+}$ binding sites (S$_{CD1}$ and S$_{CD2}$) in the CTD (Supplementary Fig. 13c). hZnT8 contains two cysteines near the C-terminus, and a unique HCH (His52-Cys53-His54) motif in a non-conserved N-terminal extension of

the neighboring protomer, whereas bacterial YiiP forms a binuclear zinc coordination with the (HHD)$_2$ motif at the CTD dimer interface (Supplementary Fig. 17c). These Zn$^{2+}$-binding residues in the CTDs of bacterial YiiP and hZnT8 are replaced by other residues in hZnT7 (Supplementary Fig. 18). Despite the absence of Zn$^{2+}$ binding sites in its CTD, hZnT7 forms a stable dimer. Thus, it seems that Zn$^{2+}$ ions bound to the TMD-CTD interface and CTD are not always necessary for the dimerization of the CDF superfamily.

### Structural and functional roles of the histidine-rich loop of hZnT7

The His-loop of hZnT7, positioned between TM4 and TM5 on the cytosolic side, is exceptionally long compared with other zinc transporters (Fig. 1). The cryo-EM density map at 2.2 Å resolution allowed for visualization of only the N-terminal (residues 229-232) and C-terminal (residues 159-163) segments of the His-loop near the ends of TM4 and TM5, respectively (Fig. 2e), suggesting that the major portion of the His-loop is highly flexible. To explore the structural and functional roles of the His-loop, its major portion (residues His176 to Pro221) was deleted, and the resultant mutant protein (hZnT7ΔHis-loop) was used for structural and functional analyses. Cryo-EM structures of the hZnT7ΔHis-loop-Fab complex in Zn$^{2+}$-bound and -unbound states were determined at nominal resolutions of 3.4 Å (Supplementary Fig. 19 and Supplementary Table 2). Since the major class (Class 4; 67.4%; 127,941 particles) showed unusually bent TM helices and a poor density map, the second major-class density map (32.6%; 69,877 particles) was used for the model building of hZnT7ΔHis-loop (Supplementary Fig. 19).

Both Zn$^{2+}$-bound and -unbound forms of hZnT7ΔHis-loop were found to adopt an OF conformation with similar location and orientation of the TM helices to those in wild-type hZnT7 (hZnT7 WT), indicating that deletion of the His-loop had little impact on the overall TMD of hZnT7 (Supplementary Fig. 20a-c). Similarly, the CTD of hZnT7ΔHis-loop is almost superimposable to that of hZnT7 WT, with an RMSD for all C$_\alpha$ atoms in this domain of 0.3 Å (Supplementary Fig. 20d). The structure of the CTD is almost insensitive to Zn$^{2+}$, displaying an RMSD for all C$_\alpha$ atoms in this domain of 0.3 Å between the Zn$^{2+}$-unbound and -bound states (Supplementary Fig. 20e). Collectively, while the His-loop may possibly contribute to the stabilization of the IF conformation via its incorporation into the widened cytosolic cavity seen in the IF protomer of the Zn$^{2+}$ state 1 (Fig. 5e), it does not appreciably determine the structure of the OF protomer.

### Essentiality of histidine residues in the His-loop for efficient Zn$^{2+}$ transport

The amino acid sequence (Fig. 1) indicates that the distribution of histidine residues in the His-loop can be classified into two sites: one near the C-terminus of TM4 (site 1 with 7 histidine residues including His164) and the other in the C-terminal half of the His-loop (site 2 with 14 histidine residues). To characterize functional roles of the sites-1 and 2 in Zn$^{2+}$ binding and Zn$^{2+}$ uptake activity, we prepared three sorts of ZnT7 mutants (HS1, HS2 and HS3). HS1 and HS2 have His-to-Ser

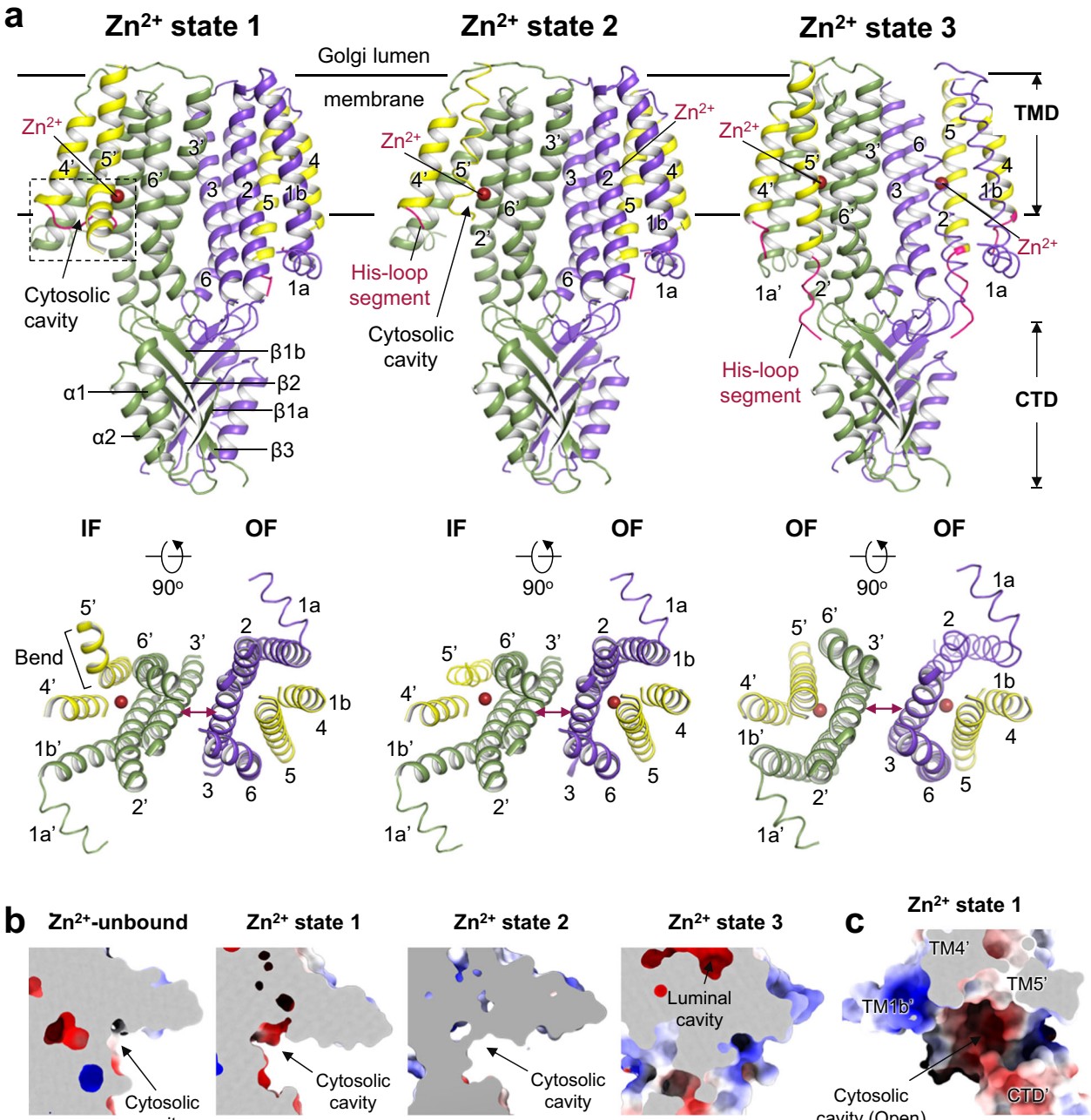

**Fig. 4 | Cryo-EM structures of the Zn²⁺-bound hZnT7 dimers (Zn²⁺ state 1, Zn²⁺ state 2 and Zn²⁺ state 3). a** Cartoon representations show the overall structures of two types of Zn²⁺-bound hZnT7 heterodimers named Zn²⁺ state 1 (left) and Zn²⁺ state 2 (middle), and a Zn²⁺-bound hZnT7 homodimer named Zn²⁺ state 3 (right). The side (upper) and cytosolic (lower) views are shown. The black dashed-line square indicates the region that is highlighted in Fig. 5e. **b** Coronal sections of the electrostatic potential surfaces of the hZnT7 dimers in Zn²⁺-unbound state, Zn²⁺ state 1, Zn²⁺ state 2 and Zn²⁺ state 3. **c** Electrostatic potential surface around the cytosolic cavity of the IF protomer in the Zn²⁺ state 1. Surface colors indicate Coulombic potentials (red, negative; white, neutral; blue, positive).

mutations at site 1 and site 2, respectively, while all histidine residues in the His-loop are mutated to Ser in HS3 (Fig. 7a). Zn²⁺ binding affinity and stoichiometry for hZnT7 WT and HS1-HS3 were analyzed by isothermal titration calorimetry (ITC) at 20 °C, pH 7.5. A set of ITC thermograms (Fig. 7b) revealed different Zn²⁺-binding properties of these four proteins. We found that each protomer of hZnT7 WT bound 3 molar equivalents of Zn²⁺ with an apparent $K_d$ value of 11.3 μM (Fig. 7c). A protomer of hZnT7 AAAA, in which the HDHD motif at the TM Zn²⁺-binding site was all mutated to alanine, bound ~2 molar equivalents of Zn²⁺ with an apparent $K_d$ value of 12.2 μM (Fig. 7c), suggesting the presence of one Zn²⁺-binding site in the TM domain and two in the His-loop of hZnT7. Each protomer of HS1 and HS2 was found to bind 2.4

and 1.7 molar equivalents of Zn²⁺ with apparent $K_d$ values of 8.8 and 11.6 μM, respectively (Fig. 7c), Altogether, it is surmised that one Zn²⁺-binding site is contained in the site 2 of the His-loop and that another Zn²⁺-binding site is constituted by both sites-1 and 2 while the site-1 or site-2 alone can bind the Zn²⁺ partially. HS3 could bind only one Zn²⁺ most likely in the TM domain.

The thermodynamic parameters, enthalpy ($\Delta H$) and entropy changes ($T\Delta S$), obtained by the ITC analysis (Fig. 7c) suggest that Zn²⁺-binding to hZnT7 is enthalpy-driven while Zn²⁺-binding to the His-loop reduced the entropy due to the decrease of conformational freedom associated with Zn²⁺ binding. Notably, the binding enthalpy for hZnT7 HS3 (−8.0 kcal/mol) greatly decreased compared to those of hZnT7

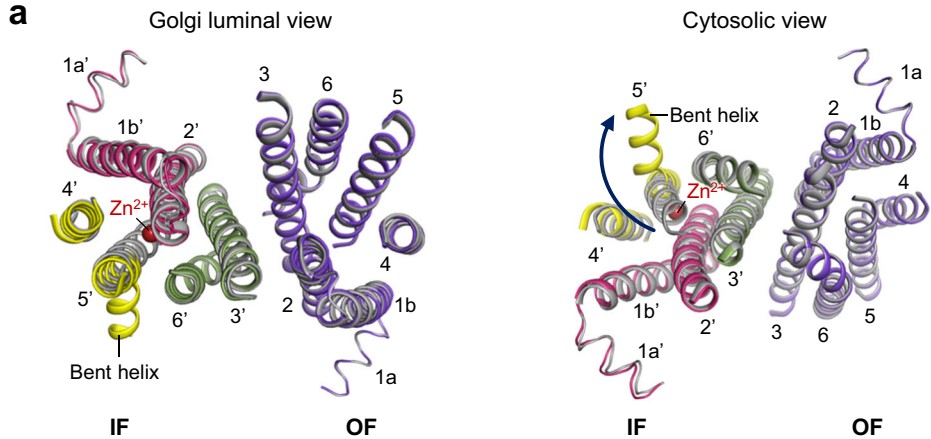

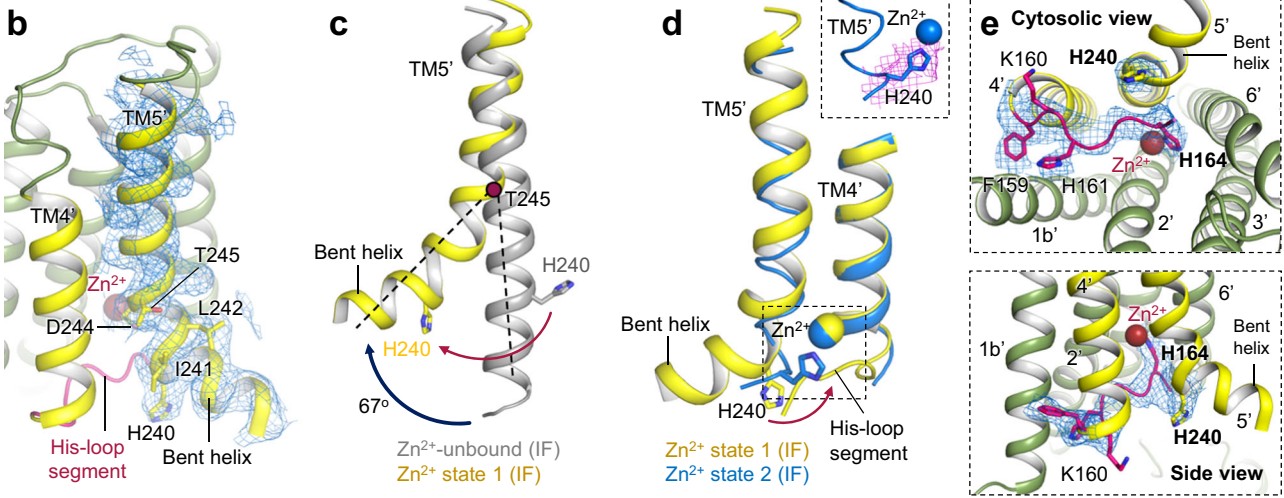

**Fig. 5 | The His-rich loop incorporates into the cytosolic cavity of the IF protomer. a** Superimposition of the TMDs of the hZnT7 heterodimers in IF-OF form (gray) and Zn²⁺ state 1 (colored as indicated), viewed from the Golgi lumen (left) and cytosol (right). Black arrows indicate the movement of TM5 during the conversion from the IF-OF form to Zn²⁺ state 1. **b** Bending of the N-terminal portion of TM5 in the IF protomer of Zn²⁺ state 1. The density map of TM5 is shown by blue mesh at a contour level of 0.17σ. **c** Superimposition of TM5 in the IF protomers of the hZnT7 heterodimers in IF-OF form (gray) and Zn²⁺ state 1 (yellow). Black and red arrows indicate the movements of TM5 and His240 in TM5 during the transition from the Zn²⁺-unbound state to Zn²⁺ state 1, respectively. **d** Superimposition of TM4 and TM5

between the IF protomers in the Zn²⁺ state 1 (yellow) and Zn²⁺ state 2 (blue). The yellow and blue spheres indicate an Zn²⁺ ion bound to the Zn²⁺ state 1 and Zn²⁺ state 2, respectively. The red arrow indicates the movement of His240 during the transition from the Zn²⁺ state 1 to state 2. The black dash-line square is highlighted in the right upper inset with the density map around His240 in the Zn²⁺ state 2. **e** Highlighted representation of the His-loop segment viewed from the cytosol (upper) and side (lower). The His-loop segment is shown in magenta with its density map (blue) at a contour level of 0.17σ. TM4 and TM5 were shown in yellow. A red sphere indicates a Zn²⁺ ion coordinated by His164.

WT, HS1 and HS2 (-15 to -16 kcal/mol), suggesting that histidine residues in the His-loop contribute to stabilizing the Zn²⁺-bound conformation of hZnT7. Consistently, the $K_d$ for Zn²⁺ value of HS3 is significantly higher (i.e. lower affinity) than those of WT, HS1 and HS2 (Fig. 7c).

To explore the roles of the His-loop histidine residues for efficient Zn²⁺ transport by hZnT7, we measured the Zn²⁺ uptake activities of HS1-HS3 by preparing the proteoliposome containing one of the mutants and Fluozin-3, and compared them with that of WT. In line with the present observation that HS1 and HS2 retained Zn²⁺-binding affinity despite lower stoichiometry, their Zn²⁺ uptake rates were comparable to or slightly lower than that of WT (Fig. 7d and Supplementary fig. 21). The Zn²⁺ uptake activity of HS3 was much lower than those of the other three. Given that the His-loop directly participates in Zn²⁺ recruitment in the IF protomer (Fig. 5d, e), one likely interpretation is that Zn²⁺ binding to the HDHD motif is promoted by the His-loop. However, HS1,

which includes a mutation of H164S, still retained the Zn²⁺ transport activity (Fig. 7d), tempting to speculate that one or several of the site-2 histidine residues can also serve as a substitute for His164. It is also notable that the ITC profile for Zn²⁺ binding to hZnT7 WT did not show stepwise Zn²⁺ bindings to different sites with different affinities. Thus, we surmise that histidine residues in the His-loop cooperate with the Zn²⁺-binding residues in the TMD for efficient Zn²⁺ recruitment by hZnT7.

## Discussion

We herein present near-atomic resolution cryo-EM structures of hZnT7 in both Zn²⁺-unbound and -bound states, in complex with the Fab fragment that specifically binds the cytosolic end of hZnT7. The cryo-EM structures illuminate multiple sub-state conformations of the hZnT7 dimer, in which two protomers adopt homodimers composed of two OF protomers and heterodimers composed of IF and OF

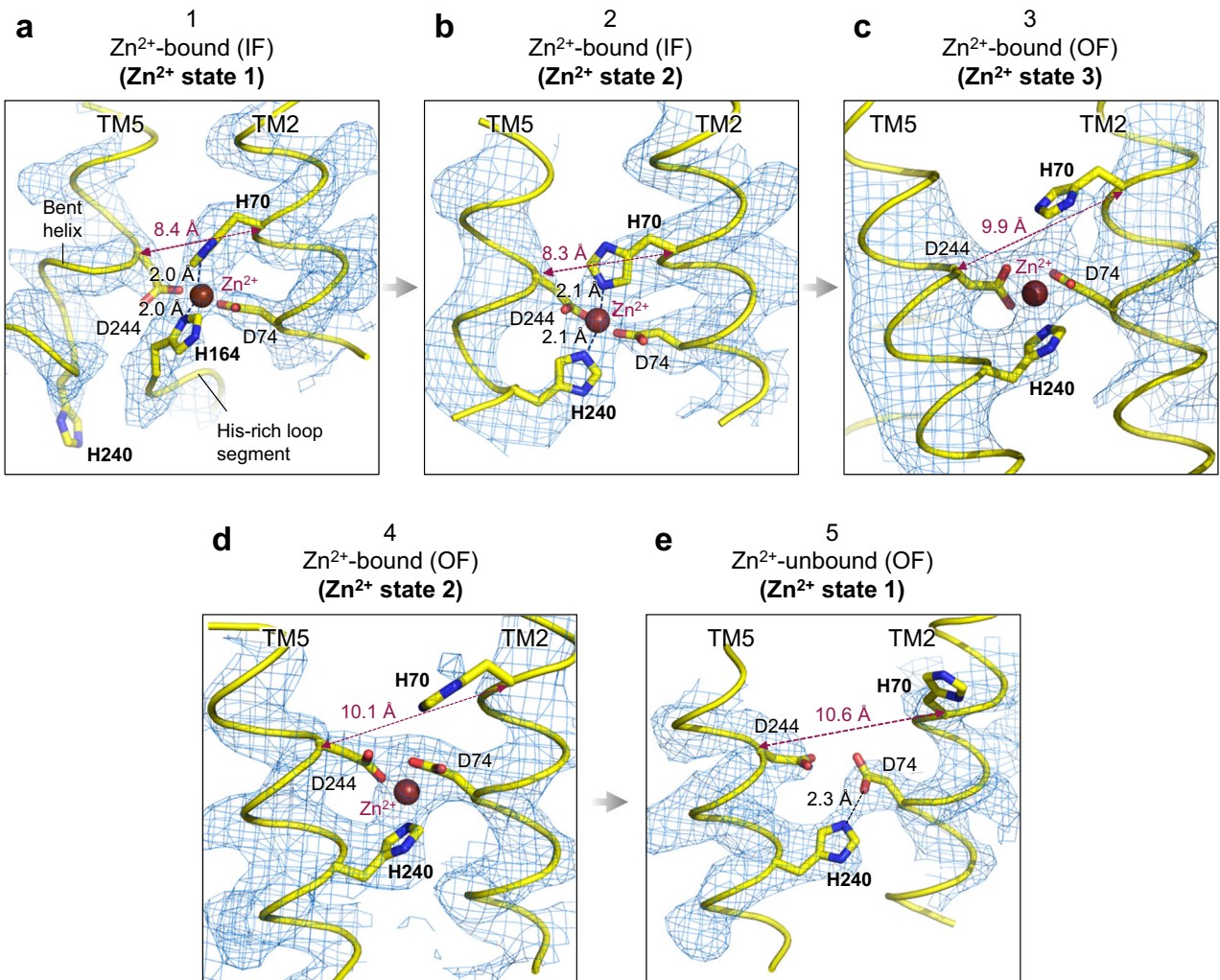

**Fig. 6 | Zn²⁺-coordination structure of hZnT7.** Close-up views of the Zn²⁺-binding sites in the IF protomer in Zn²⁺ state 1 (**a**) and Zn²⁺ state 2 (**b**), in the OF protomer in Zn²⁺ state 3 (**c**), Zn²⁺ state 2 (**d**) and Zn²⁺ state 1 (**e**). The red spheres indicate bound Zn²⁺. The density map and Zn²⁺-binding residues are represented by blue mesh and yellow sticks, respectively. The main chains are represented by yellow ribbons. Red dashed lines indicate the distance between the Cα atoms of His70 and Asp244. Black dashed lines indicate the distances between the Nδ/Nε atom of His70, His164 or His240 and Zn²⁺.

protomers regardless of Zn²⁺ binding. In a Zn²⁺-bound IF protomer of Zn²⁺ state 1, an N-terminal His-loop segment is inserted into the cytosolic cavity via the bending of the cytosolic half of TM5, allowing His164 to directly coordinate to Zn²⁺. In another Zn²⁺-bound IF protomer of Zn²⁺ state 2, His240 engages in Zn²⁺ coordination at the TM Zn²⁺-binding site in place of His164. Zn²⁺ state 1 was identified as the major subclass conformation when using the sample purified throughout in the presence of 10 μM Zn²⁺, whereas Zn²⁺ state 2 was observed after the exogenous addition of 200 μM or 300 μM Zn²⁺ to the hZnT7. Indeed, our ITC analysis showed that the $K_d$ for Zn²⁺ value of purified hZnT7 is around 10 μM (Fig. 7). However, Outten & O'Halloran (2001) estimated the labile Zn²⁺ concentration in the cytosol to be in the picomolar range, much below 10 μM[20]. More recently, Liu et at. (2022) reported that the cytosolic labile Zn²⁺ concentration in Hela cells is ~0.13 nM[21]. Thus, we need to carefully consider the discrepancy between the Zn²⁺-binding affinity of hZnT7 determined by the in vitro biophysical approach and the Zn²⁺ availability in the physiological environment as a central enigma in mechanisms of zinc transporters. Nonetheless, the different His-loop conformations seen at the different Zn²⁺ concentrations may suggest that low concentrations of Zn²⁺ allow the His-loop to efficiently transfer Zn²⁺ to the Zn²⁺-binding site in the TMD, whereas the large excess of Zn²⁺ could limit the mobility of the His-loop possibly via multiple Zn²⁺ binding, resulting in trapping the Zn²⁺ state 2

with the His-loop located away from the TM Zn²⁺-binding site. No homogeneous IF-IF dimer was identified in either Zn²⁺-bound or unbound form, implying that this state may be energetically unstable as suggested by the well-separated locations of TM2 of one protomer and TM3 of another protomer in the modelled IF-IF homodimer (Supplementary Fig. 22).

Like hZnT7, an *Arabidopsis thaliana* vacuolar Zn²⁺/H⁺ antiporter (AtMTP1) also has a long cytosolic loop with 25 histidine residues, which can be divided into the 1ˢᵗ (near the C-terminus of TM4) and 2ⁿᵈ (central part) histidine-rich segments[22,23]. The ITC analysis showed that a polypeptide chain of the AtMTP1 His-rich loop was able to bind four Zn²⁺ ions[20]. Nonetheless, the deletion of the N-terminal 32 amino acid residues in the His-rich loop led to much higher Zn²⁺ uptake activity than the wild type[23]. Thus, the His-rich loop may rather serve to suppress the Zn²⁺ transport activity of AtMTP1, presumably to maintain intracellular Zn²⁺ homeostasis in accordance with the Zn²⁺ concentration. Molecular mechanisms underlying the opposite functional roles of the His-loop between hZnT7 and AtMTP1 need to be further explored.

The bending of TM5 in hZnT7 is caused by the kink at Thr245, leading to the large movements of His240 (Fig. 5c). As a result, the conventional Zn²⁺-binding HDHD motif is partially affected by the transient coordination of His164 to Zn²⁺. Similar TM helix bending can

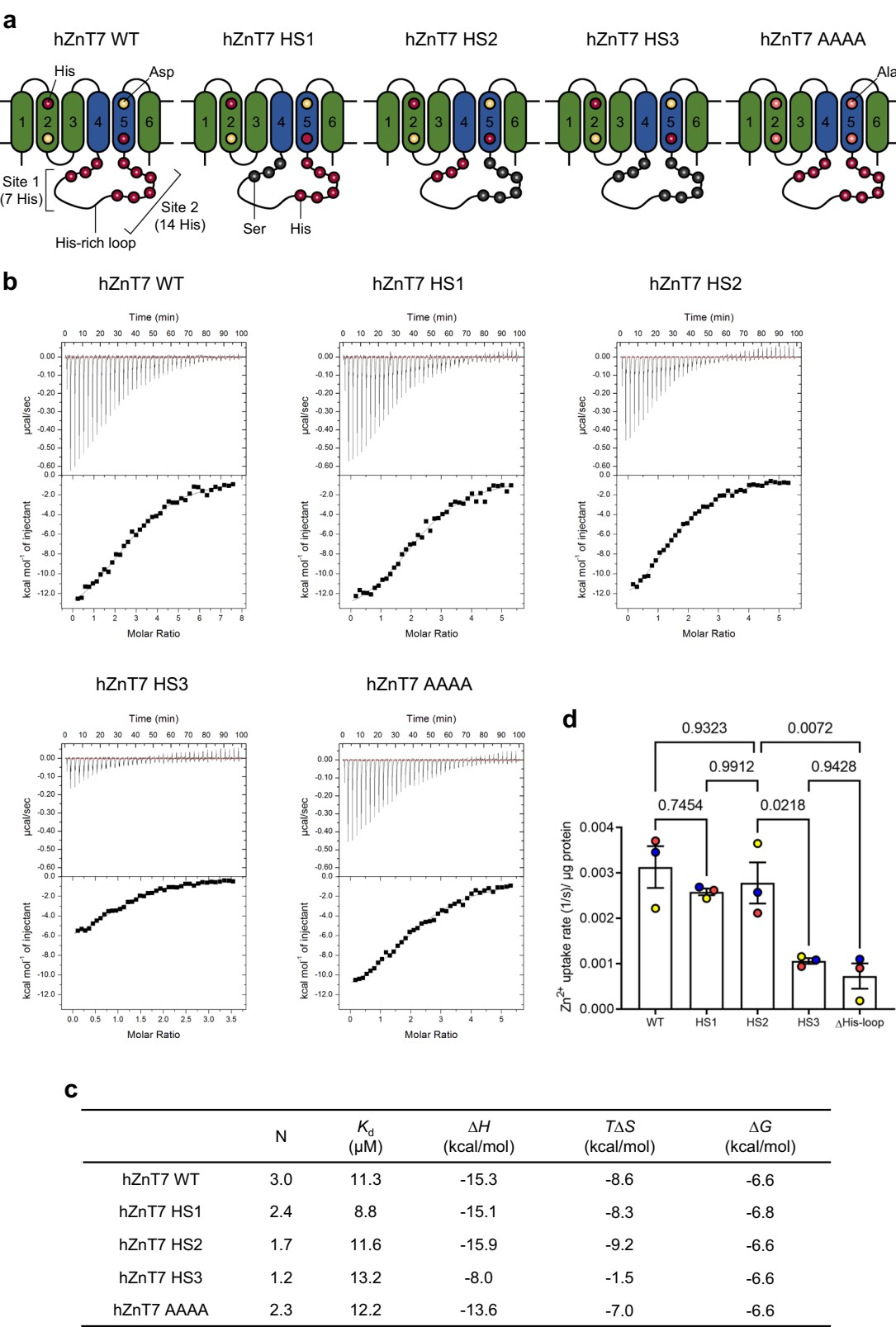

be seen in other transporters of known structures, including *Pyrococcus furiosus* MATE (PfMATE)[24] and *Arabidopsis thaliana* DTX14 (AtDTX14)[25] that belong to the MATE family transporters responsible for efflux of endogenous cationic, lipophilic substances, and xenobiotics. In PfMATE[24], the kink at Pro26 is somehow remotely induced by the protonation of Asp23 on the N-terminal lobe of TM1, leading to the bending of this helix. A similar event has been observed in

AtDTX14[25], where Glu265 (TM1) at the substrate-binding site is protonated, inducing the kink of TM1 at Pro257. Although such a critical proline residue does not exist near Asp244 in TM5 of hZnT7, the deprotonation/protonation of Asp244 may possibly control the bending of TM5. In any case, $Zn^{2+}$ binding that induces the incorporation of the His-loop into the cytosolic cavity seems likely to be closely coupled to the bending of TM5 in hZnT7.

**Fig. 7 | Roles of the His-loop in Zn²⁺ binding affinity and Zn²⁺ uptake activity of hZnT7. a** Cartoon representations show the distribution of histidine residues in the His-loop, and the constructions of HS1, HS2, HS3 and AAAA mutants of hZnT7. **b** ITC analyses for Zn²⁺ binding to hZnT7 WT and its His-to-Ser mutants (HS1, HS2 and HS3) and AAAA mutant with mutations of the HDHD motif to alanine. The upper panels show the heat generation during the calorimetric titrations with 1 μL injections of 500 μM ZnCl₂ into hZnT7 WT (18 μM), HS1 (20 μM), HS2 (20 μM), HS3 (30 μM) and AAAA (20 μM) in the reaction cell with a volume of 280 μL. The lower panels display the integrated heat values calculated from the upper panels as a function of the molar ratio of Zn²⁺ to hZnT7. The solid lines represent the best fit to the observed ITC curves. At least two independent replicates for each construct. **c** The resulting fitting parameters are summarized. **d** Initial Zn²⁺ uptake rates per μg of hZnT7 protein determined with the proteoliposomes containing the indicated ZnT7 protein and Fluozin3 in buffer containing 100 μM Zn²⁺. Data are averages from three independent experiments. Error-bars indicate mean ± SEM (*n* = 3); numbers indicate *p* values analyzed by one-way ANOVA. Source numerical data are provided in the Source Data file. See also Supplementary Fig. 21.

The present cryo-EM analysis also demonstrates that the Zn²⁺-bound OF form has structural features suitable for Zn²⁺ release to the Golgi lumen. The luminal cavity of the Zn²⁺ transport pore is open, whereas the cytosolic cavity is fully closed to prevent further Zn²⁺ entry from the cytosol. Presumably, H⁺ from the weakly acidic Golgi lumen protonates residues of the HDHD motif, triggering the release of bound Zn²⁺. The negatively charged surface of the luminal cavity (Fig. 2d) likely facilitates Zn²⁺ transfer to the luminal gate in the OF form. Protonations at the HDHD motif may also induce the subsequent conversion from the OF to the IF form, followed by the release of H⁺ into the cytosol. This mechanism of action nicely explains the physiological function of hZnT7 as a Zn²⁺/H⁺ antiporter in the Golgi membrane.

Taken all together, we propose a likely scenario of hZnT7-mediated Zn²⁺ transport from the cytosol to the Golgi lumen (Fig. 8 and Supplementary Movie 3). The Zn²⁺-unbound IF protomer possesses a negatively charged cytosolic cavity around the TMD-CTD interface, near which the exceptionally long and flexible His-loop is located (stage 1). Upon Zn²⁺-binding, an N-terminal segment of the His-loop is incorporated into the widely opened cytosolic cavity via the bending of the cytosolic half of TM5, where His164 in the His-loop directly and transiently coordinates to Zn²⁺ to form a tetrahedral complex with His70, Asp74 and Asp244 in the TMD (stage 2). This conformational change seems advantageous for the efficient recruitment and shuttling of Zn²⁺ to the metal transport pathway in the IF protomer. The next step is the exchange of a Zn²⁺-coordinating residue from His164 to His240 (TM5) via the return of the bent segment of TM5 (stage 3). This state likely allows the conversion from the IF to OF conformations, concomitant with the movement of His70 away from the bound Zn²⁺ possibly via protonation to the Zn²⁺-binding residues (stages 4 and 5). Finally, Zn²⁺ is released to the Golgi lumen (stage 6).

While the present study has significantly advanced our understanding of the Zn²⁺ transport mechanism exerted by an hZnT7 protomer, it remains an important open question how two protomers work cooperatively within the hZnT7 dimer during the Zn²⁺-transport cycle. It is also possible that two protomers work independently. Further exploration of other intermediate states of the hZnT7 dimer by advanced structural approaches including the time-resolved cryo-EM single-particle analysis[26] may reveal a whole picture of the hZnT7-mediated Zn²⁺ transport. ZnT7 is likely to collaborate with other ZnT and ZiP family members for the maintenance of Zn²⁺ homeostasis in the early secretory pathway. Detailed mechanisms of intracellular Zn²⁺ homeostasis resulting from the cooperation of multiple Zn²⁺ transporters is undoubtedly an important topics of the present-day molecular cell biology, in light of the physiological roles of Zn²⁺ in the folding, maturation, and quality control of Zn²⁺-binding proteins, cell signaling and regulation, and many other cellular events.

## Methods
### Cloning and protein expression
A cDNA encoding human ZnT7 (hZnT7; UniPort ID: Q8NEW0) was synthesized with codon optimization for expression in human cells. The hZnT7 gene with an N-terminal PA-tag sequence (GVAMP-GAEDDVV) was cloned into a PiggyBac Cumate Switch Inducible Vector (System Biosciences). For large-scale expression of hZnT7, stable

PiggyBac-hZnT7-expressing HEK293T cells were cultured in DMEM (High glucose; 4.5 g/L D-(+)-glucose with L-glutamine, phenol red and sodium pyruvate; Nacalai Tesque, Cat# 08458-16), supplemented with 4% fetal bovine serum (FBS, Nichirei Biosciences, Cat# 175012), 1% Penicillin-Streptomycin mixed solution (PS, Nacalai Tesque, Cat# 26253-84) and 4 μg/mL Puromycin (InvivoGen, Cat# ant-pr-1), and grown at 37 °C in an atmosphere containing 5% CO₂. After 3 days, the cells were passaged into fresh DMEM (high glucose) supplemented with 4% FBS and 1% PS. After growth for 24 h, expression was induced by the addition of 4-Isopropylbenzoic acid (Cumate, Wako, Cat# 039-06411) and Phorbol 12-Myristate 13-Acetate (PMA, Wako, Cat# 162-23591) to final concentrations of 10X (300 μg/mL) and 1X (50 nM), respectively. Cells were continuously cultured at 37 °C for 3 h in 5% CO₂ before being incubated at 30 °C in 5% CO₂. After 48 h, the cells were harvested, washed with cold-1X phosphate-buffered saline (PBS), and stored at -80 °C.

To prepare the hZnT7ΔHis-loop, the DNA segment encoding residues His176 to Pro221 of hZnT7 was deleted using the primers as shown in Supplementary Data 1. PCR product was amplified with PrimeSTAR® Max DNA Polymerase (Takara, Cat# R045Q) and ligated by In-Fusion Cloning (In-Fusion HD kit, Takara, Cat# ST2317). All primer and double-strand DNA in this study were synthesized by Eurofins. The medium was changed to fresh serum-free DMEM containing 1% PS, and HEK293T cells (ATCC, Cat# CRL-3216) were transfected with pcDNA3.1-PA-hZnT7ΔHis-loop plasmids using PEI reagent (Sigma-Aldrich, Cat# 408727-100 ML) at a ratio of 50 μg DNA to 150 μL PEI reagent (1 mg/mL). Cells were continuously cultured for 24 h before 10 mM sodium butyrate (Nacalai Tesque, Cat# 12328-44) was added. After 54 h of culturing at 30 °C and 5% CO₂, the cells were harvested.

To prepare the hZnT7 HS1, HS2 and HS3 mutants, histidine residues in the His-loop were mutated to serine. For hZnT7 HS1, 7 histidine residues near the C-terminus of TM4 (all histidine residue numbers, e.g. His161, 164, 166, 168, 172, 174 and 176) were mutated to serine. For hZnT7 HS2, 14 histidine residues near the N-terminus of TM5 (all histidine residue numbers, e.g. His187, 189, 192, 194, 196, 200, 204, 206, 208, 210, 212, 214, 216 and 218) were mutated to serine. In hZnT7 HS3, all histidine residues raised above were mutated to serine. Primers and double-strand DNA were shown in Supplementary Data 1. For hZnT7 AAAA mutant, His70 and Asp74 (TM2), and His240 and Asp244 (TM5) were mutated to alanine.

### Protein purification
Proteins were purified and treated at 4 °C. Cells were washed with ice-cold-1X PBS buffer and collected by centrifugation. Cells from 2 L of medium (wet-weight, ~8 g) were lysed by resuspension in buffer A containing 20 mM Tris-HCl (Nacalai Tesque, Cat# 35434-34), pH 7.5, 150 mM NaCl (Nacalai Tesque, Cat# 31320-05) and 10% glycerol (Nacalai Tesque, Cat# 17018-83), supplemented with 2 mM DTT (Nacalai Tesque, Cat# 14112-94), 0.05 mg/mL DNase I (Wako, Cat# 043-26773), 5 mM MgCl₂ (Nacalai Tesque, Cat# 20908-65), and 1× protease inhibitor cocktail (Nacalai Tesque, Cat# 25955-11). The cells were sonicated for 3 min in an ice-water cup, and the cell membrane fraction was collected by ultracentrifugation at 200,000 × g (micro Ultracentrifuge CS100FNX, Hitachi) for 1 h at 4 °C. The pellet was dissolved in a solution containing 1% (v/v) n-dodecyl-β-D-maltoside (DDM,

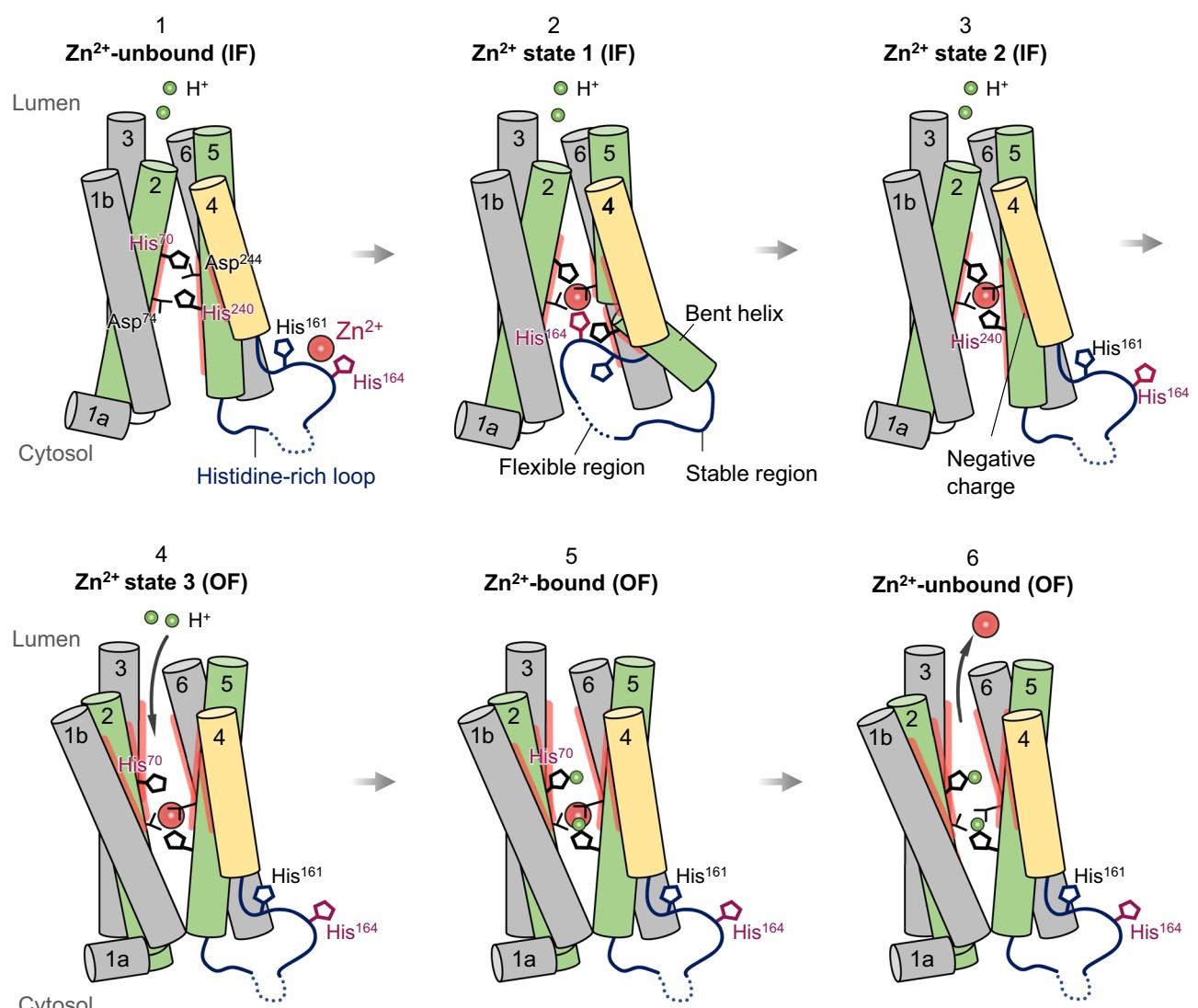

**Fig. 8 | Proposed model of $Zn^{2+}$ transport mediated by a protomer of Golgi-resident $Zn^{2+}$/H+ antiporter hZnT7.** During its $Zn^{2+}$ transport cycle, an hZnT7 protomer undergoes TM-helix rearrangement and striking His-loop movement, leading to the facilitated $Zn^{2+}$ recruitment from the cytosol to the negatively charged cytosolic cavity. During this process, the exchange of a $Zn^{2+}$-coordinating residue from His164 to His240 takes place. Once $Zn^{2+}$ is bound to the HDHD motif, the IF form converts to the OF form, concomitant with moving away of His70. In the meantime, protons likely enter the luminal cavity to protonate His70 and possibly His240. These conformational changes around the $Zn^{2+}$-binding site likely serve to facilitate the release of $Zn^{2+}$ to the luminal side. The TM helices are presented as tubes; TM1a, TM1b, TM3 and TM6 in gray, TM2 and TM5 in green, and TM4 in yellow. The visible His-loop segment around the C-terminus of TM4 and N-terminus of TM5 is represented by dark blue lines while dashed lines indicate the invisible region of the His-loop. $Zn^{2+}$ is represented by a red sphere. The negatively charged surfaces of the cytosolic cavity in the IF conformation and the $Zn^{2+}$ release gate in the OF conformation are represented by thick red lines. A proton (H+) is represented by a green sphere. Note that the CTD, TM loops and another protomer of the hZnT7 homodimer are omitted for simplification. How two protomers interplay each other within the hZnT7 dimer during the $Zn^{2+}$-transport cycle remains elusive. See also Supplementary Fig. 23.

Nacalai Tesque, Cat# 14239-54), 0.5% (v/v) cholesteryl hemisuccinate (CHS, Sigma-Aldrich, Cat# C6512-5G), 20 mM Tris-HCl, pH 7.5, 150 mM NaCl, 10% glycerol and 2 mM DTT, and incubated for 2 h at 4 °C with constant stirring. The supernatant containing PA-tagged hZnT7 was collected by centrifugation at 15,000 × g for 30 min at 4 °C and incubated with 12 ml anti-PA-tag antibody beads (net weight, 6 ml; Wako, Cat# 018-25843) overnight at 4 °C with gentle rotation. The resin was collected in a glass column (Bio-Rad) and washed with 15 column volume (CV) of Buffer B [Buffer A supplemented with 0.02% GDN (Anatrace, Cat# GDN101-1G)]. PA-tagged hZnT7 was eluted with Buffer C [Buffer B supplemented with 0.2 mg/mL PA peptide (PH Japan Co., Ltd; Cat# 180688)]. The protein was concentrated using a 50-kDa cutoff Amicon filter (Sigma-Aldrich) and purified on a Superose™ 6 Increase 10/300 GL gel filtration column (GE Healthcare) pre-equilibrated with buffer B. The eluted fractions were concentrated

using a 50-kDa cutoff Amicon filter and mixed with an excess amount of the Fab#1 fragment. The hZnT7-Fab#1 mixture was further subjected to Superdex™ 200 Increase 10/300 GL gel filtration column (GE Healthcare) pre-equilibrated with glycerol-free buffer B to remove the residual Fab#1 fragments and concentrated to ~10 mg/mL using a 100-kDa cutoff Amicon filter (Sigma-Aldrich) for negative staining EM and cryo-EM measurements. To obtain higher resolution cryo-EM maps, purification of $Zn^{2+}$-unbound hZnT7 was performed throughout in the presence of EDTA.

hZnT7ΔHis-loop, HS1, HS2, HS3 and AAAA mutants were purified as described for wild-type hZnT7 (hZnT7 WT). The elution fractions from the affinity chromatography column were concentrated using a 50-kDa cutoff Amicon filter and then incubated with 1 mM EDTA (Nacalai Tesque, Cat# 15105-35) for 30 min on ice before being purified on a Superose™ 6 Increase 10/300 GL gel filtration column pre-

equilibrated with buffer B (20 mM Tris-HCl, pH 7.5, 150 mM NaCl and 0.02% GDN). The eluted fractions were concentrated using a 50-kDa-cutoff Amicon filter and used for the ITC analysis. For cryo-EM analysis, hZnT7ΔHis-loop protein was eluted from the anti-PA-tag antibody beads, concentrated using a 50-kDa cutoff Amicon filter, and purified on a Superose™ 6 Increase 10/300 GL gel filtration column pre-equilibrated with buffer B. The eluted fractions were again concentrated using a 50-kDa-cutoff Amicon filter and used for preparation of the hZnT7ΔHis-loop-Fab#1 complex in the same way as the hZnT7 WT-Fab#1 complex.

### Preparation and screening of monoclonal antibody Fab fragments to hZnT7

All animal experiments conformed to the guidelines of the Guide for the Care and Use of Laboratory Animals of Japan and were approved by the Kyoto University Animal Experimentation Committee. Mouse monoclonal antibodies against hZnT7 were generated essentially as described[27]. Briefly, a proteoliposome antigen was prepared by reconstituting purified hZnT7 at high density into phospholipid vesicles consisting of a 10:1 mixture of chicken egg yolk phosphatidylcholine (egg PC, Avanti Polar Lipids) and adjuvant lipid A (Sigma-Aldrich) to facilitate the immune response. 6-weeks old female MRL/lpr mice maintained at temperature and humidity ranges of 22 to 26 °C and 40% to 60%, respectively, under a 12-h light and 12-h dark cycle were injected three times at 2-week intervals with the proteoliposome antigen. Antibody-producing hybridoma cell lines were generated using a conventional fusion protocol. Biotinylated proteoliposomes were prepared by reconstituting hZnT7 with a mixture of egg PC and 1,2-dipal-mitoyl-sn-glycero-3-phosphoethanolamine-N-(cap biotinyl) (16:0 biotinyl Cap-PE; Avanti), and used as binding targets for conformation-specific antibody selection. The targets were immobilized onto streptavidin-coated microplates (Nunc). Hybridoma clones producing antibodies recognizing conformational epitopes of hZnT7 were selected by enzyme-linked immunosorbent assay on immobilized biotinylated proteoliposomes (liposome ELISA), allowing positive selection of the antibodies that recognized the native conformation of hZnT7. Serum was diluted in the range of 3/10,000,000 to 1/1,000, and 1-10 μg/mL of the hZnT7 antigen was used for ELISA experiments. Clones were additionally screened for reduced antibody binding to SDS-denatured hZnT7, resulting in negative selection of linear epitope-recognizing antibodies. The formation of stable complexes between hZnT7 and each antibody clone was determined by fluorescence-detection SEC. Five monoclonal antibodies were found to specifically bind to and stabilize conformational epitopes of hZnT7. The sequences of the Fabs were determined by standard 5′-RACE using total RNA isolated from hybridoma cells (Supplementary Data 2).

### Preparation of hZnT7-Fab complexes

Purified hZnT7 was mixed at a molar ratio of 1:3 with each of the five selected Fab fragments (Fab#1, YN7114-08; Fab#2, YN7117-01; Fab#3, YN7114-03; Fab#4, YN-7148-12 and Fab#5, YN7179-03). Following incubation for 30 min at 4 °C with gentle rotation, the mixtures were centrifuged at 15,000 rpm for 10 min at 4 °C and injected into a Superdex™ 200 Increase 10/300 GL gel filtration column. Fractions containing both hZnT7 and Fab were collected and concentrated with a 100-kDa cutoff Amicon filter to around 10 mg/mL for cryo-EM measurements.

To prepare $Zn^{2+}$-bound hZnT7, hZnT7 purified in $Zn^{2+}$-free buffer was mixed with 10 μM $Zn^{2+}$ and incubated on ice for 10 min. This mixture was incubated with a 3-molar excess of Fab#1 for 0.5 h at 4 °C with gentle rotation, centrifuged at high speed to remove precipitates, and injected onto an SEC column, as described above, with the SEC buffer containing 10 μM $Zn^{2+}$. Fractions containing both hZnT7 and Fab#1 were pooled and concentrated using a 100-kDa cutoff Amicon filter to ~8-10 mg/mL for cryo-EM measurements.

### Negative stain EM

The hZnT7-Fab complexes were assessed by negative-stain EM, which provided 2D reconstruction images useful for Fab selection. Grids for negative-stain EM were prepared according to the standard protocol. Specifically, 5 μL of purified hZnT7-Fab complex was applied to glow-discharged carbon-coated grids (ELC-C10), with the excess solution removed with filter paper. The grid was immediately stained with 1% (w/v) uranyl acetate solution. The negatively stained EM grids were imaged on a JEOL JEM-2010F electron microscope (JEOL) operated at 200 kV. Particles were extracted by phase-flipping, followed by reference-free 2D class averaging using RELION v3.1[28,29].

### EM data acquisition

For the initial data collection using a Talos Arctica TEM (ThermoFisher Scientific), 3 μL of the hZnT7-Fab#1 or #3 complex samples were applied to glow-discharged QUANTIFOIL® (QUANTIFOIL® R 1.2/1.3, 300 mesh, Cu-Rh). The grids were blotted and immersed in liquid ethane using Vitrobot Mark IV systems (FEI; ThermoFisher Scientific) operated at 4 °C and 100% humidity. The blotting parameters for hZnT7 WT were set at a wait time of 3 s, a blot time of 3.5-4 s, and a blot force of 10; and the blotting parameters for hZnT7ΔHis-loop were set at a wait time of 3 s, a blot time of 3 s, and a blotting force of -5. For the final data collection using a CRYO ARM™ 300II (JEOL), the hZnT7-Fab#1 complex sample with or without $Zn^{2+}$ was applied to glow-discharged QUANTIFOIL® (QUANTIFOIL® R 1.2/1.3, 200 mesh, Au). The grids were blotted and immersed in liquid ethane using Vitrobot Mark IV systems (ThermoFisher Scientific) operated at 4 °C and 100% humidity, with a wait time of 3 s, a blot time of 3 s, and a blot force of -5. For preparation of the hZnT7-Fab#1 complex with exogenously added $Zn^{2+}$, the sample purified in the presence of 10 μM $Zn^{2+}$ was mixed with 200 or 300 μM $Zn^{2+}$ and then immediately blotted under the same condition as above.

The grids were initially imaged using a Talos Arctica TEM (ThermoFisher Scientific) equipped with a Gatan K2 summit. The final datasets of the hZnT7-Fab#1 and hZnT7ΔHis-loop-Fab#1 complexes were collected on a CRYO ARM™ 300II (JEOL) operated at 300 kV and equipped with a JEOL in-column Omega energy filter and a Gatan K3 BioQuantum detector. Data were automatically collected using SerialEM v3.7[30]. The data collection parameters are summarized in Supplementary Tables 1 and 2.

### EM data processing

Details of image processing are summarized in Supplementary Fig. 3, 13, 15 and 19. All movie stacks were subjected to motion correction using the motion correction program implemented in RELION v3.1[28,29] or v4.0[31], and the contrast transfer function (CTF) parameters were estimated with Patch CTF estimation in cryoSPARC v3.3[32] or CTFFIND4 v4.1.5[33]. In the initial data collection by using Titan Krios G3i, we determined a cryo-EM structure of the hZnT7-Fab#1 complex at 2.8 Å resolution, which was used as 2D templates for template picking in the analysis of the following datasets.

For the dataset of $Zn^{2+}$-unbound hZnT7, particles were picked by template picker in cryoSPARC from 6,280 micrographs with CTF resolution of less than 6 Å resolution, and then subjected to 2D classification. The best particles (403,333 particles) were subjected to ab-initio modeling and heterogeneous refinement. Particle belonging to the best class (272,633 particles) were re-extracted at a pixel size of 0.788 Å, and then subjected to non-uniform (NU) refinement. The refined particles were subjected to two arounds of Bayesian polishing and NU-refinement with global CTF and defocus refinement. The UCSF pyem v0.5[34] was used to data conversion from cryoSPARC to RELION. After an additional ab-initio modeling and heterogeneous refinement, particles belonging to the best two classes with high-resolution maps (183,862 particles) were subjected to NU-refinement in C2 symmetry, resulting in an EM map reconstructed at 2.07 Å resolution (consensus

map). To estimate the conformational heterogeneity of hZnT7, 3D Variability Analysis (3DVA)[35] and its clustering analysis in cryoSPARC were performed by using a mask covering the hZnT7 portion. Particles belonging to three clusters in which two protomers adopted an OF conformation (OF-OF form), with clear density of the TM regions, were selected and subjected to NU-refinement. We found that in another cluster (28,335 particles), one protomer adopted in an IF conformation and the other in an OF conformation (IF-OF form). The particles belonging to this cluster were also selected and subjected to NU-refinement in the C1 symmetry. Finally, to improve the density of the hZnT7 portion, the refined particles were subjected to local refinement using a mask covering the hZnT7 dimer, at rotation and shift standard deviation of 3º and 2 Å, respectively. The final maps of the OF-OF and IF-OF forms of $Zn^{2+}$-unbound hZnT7 were reconstructed at 2.21 and 2.79 Å-resolution, respectively (Supplementary Fig. 3).

For the dataset of $Zn^{2+}$-bound hZnT7 prepared in the presence of 10 μM $Zn^{2+}$ (named $Zn^{2+}$ sample 1), particles were picked by template picker from 6,775 micrographs, and then subjected to 2D classification. The best classes (300,742 particles) were subjected to two rounds of ab-initio modeling and heterogeneous refinement. Particles belonging to the best class (195,376 particles) were re-extracted at a pixel size of 0.788 Å, and then subjected to NU-refinement, followed by two rounds of Bayesian polishing and NU-refinement with global CTF and defocus refinement. After 3DVA, particles belonging to good clusters with clear density of the TM regions were selected and subjected to NU refinement and local refinement. The final map of the $Zn^{2+}$ state 1 (IF/Zn-OF form) was reconstructed at 2.68 Å resolution, in which one protomer adopted a $Zn^{2+}$-bound IF conformation and the other in a $Zn^{2+}$-unbound OF conformation (Supplementary Fig. 13).

For the datasets of the other $Zn^{2+}$-bound hZnT7 sample prepared by addition of 200 μM or 300 μM $Zn^{2+}$, particles were picked by template picker from 6,135 micrographs and then subjected to 2D classification. The best classes (229,309 particles) were subjected to two rounds of ab-initio modeling and heterogeneous refinement. Particles belonging to the best class (137,123 particles) were re-extracted at a pixel size of 0.788 Å, and then subjected to NU-refinement, followed by two rounds of Bayesian polishing and NU refinement with global CTF and defocus refinement. Subsequent 3DVA focused on the IF protomer of the hZnT7 portion revealed that particles were classified into two different forms, an OF/Zn-OF/Zn homodimer form (63,620 particles) and an IF/Zn-OF heterodimer form (60,692 particles). Similar results were obtained from the sample prepared by addition of 200 μM $Zn^{2+}$. Subsequently, the classes of these two forms were merged and subjected to NU-refinement, resulting in the improvement of overall density for each class. For the IF/Zn-OF/Zn form ($Zn^{2+}$ state 2), particles were further selected after additional heterogenous refinement (98,881 particles), and then subjected to NU-refinement and local refinement. The final map of the $Zn^{2+}$ state 2 was determined at 2.92 Å resolution, in which one protomer existed in a $Zn^{2+}$-bound IF conformation and the other in a $Zn^{2+}$-bound OF conformation. For the OF/Zn-OF/Zn form ($Zn^{2+}$ state 3), additional ab-initio modeling, heterogenous refinement, and 3DVA were performed. The best particles (56,678 particles) were subjected to NU-refinement and local refinement, resulting in an EM map of "$Zn^{2+}$ state 3" at 3.12 Å resolution, in which both protomer adopted a $Zn^{2+}$-bound OF conformation (Supplementary Fig. 15).

For the dataset of hZnT7ΔHis-loop in $Zn^{2+}$-free buffer, particles were picked from 1,213 movies using LoG-based auto-picking in RELION to create 2D references for template-based picking from the full dataset of 4,850 movies. After two rounds of 2D classification, 325,036 particles were selected and subjected to two rounds of 3D classification using the 3D initial model generated by RELION. Particles classified into the best class (188,918 particles) were re-extracted at a pixel size of 1.182 Å and subjected to 3D refinement in the C2 symmetry, CTF refinement, and Bayesian polishing. Subsequent

focused non-aligned 3D classification in the C2 symmetry using an encompassing mask of the TM domain improved the local resolution of the latter. The best particles with clear density for the TM domain (Class 2, 60,977 particles) were refined at 3.4 Å resolution with non-uniform-refinement in cryoSPARC (Supplementary Fig. 19).

For the dataset of the hZnT7ΔHis-loop in $Zn^{2+}$-containing buffer, particles were picked from 500 movies by using LoG-based auto-picking in RELION to create 2D references for template-based picking from the full dataset of 6,150 movies. After two rounds of 2D classification, a total of 424,349 particles were selected and subjected to two rounds of 3D classification. Particles classified into the best class (142,746 particles) were re-extracted at a pixel size of 1.182 Å and subjected to 3D refinement in C2 symmetry, CTF refinement, and Bayesian polishing. Subsequent focused non-aligned 3D classification in C2 symmetry using an encompassing mask of the TM domain improved the local resolution of the latter. The best particles (142,624 particles) were refined at 3.4 Å resolution with non-uniform refinement in cryoSPARC (Supplementary Fig. 19).

The global resolution was estimated in RELION v3.1 or v4.0 and cryoSPARC v3.3 with a Fourier shell correlation (FCS) of 0.143. The EM maps were sharpened and locally filtered based on local resolution with cryoSPARC v3.3.

## Model building, refinement and validation

An initial model of the $Zn^{2+}$-free hZnT7-Fab#1 complex was automatically built with Buccaneer in CCP-EM v1.6.0[36]. Further manual model building was performed with COOT v0.9.8.7[37,38]. The model was refined against the locally filtered map using phenix.real_space_refine in PHENIX v1.20.1[39]. The final model included hZnT7 residues 22-376, Fab light chain residues 1-218, and Fab heavy chain residues 1-234 (Supplementary Data 2). The large part of the His-loop (residues 164 to 228) was missing from the EM map. Residues 135-140 of hZnT7 were removed from the model because of poor density. Structures of the other forms of hZnT7 were initially modeled using the $Zn^{2+}$-unbound OF-OF structure, followed by further manual model building with COOT v0.9.8.7. The models were refined against the locally filtered maps using phenix.real_space_refine in PHENIX v1.20.1.

The final models were validated using MolProbity[40]. All the figures were prepared in PyMOL v2.5.2[41], UCSF Chimera v1.17[42] and UCSF ChimeraX v1.5[43]. Amino acid sequence alignments were analyzed by the MAFFT version 7 online service with defaults[44]. Electrostatic potential maps were calculated with APBS-PDB2PQR online service with pKa Options setup at pH 7.5[45,46].

## Isothermal titration calorimetry analysis

Isothermal titration calorimetry (ITC) experiments were carried out using the MicroCal™ iTC200 calorimeter (GE Healthcare). Typically, a 0.4 μL initial injection and 1.0 μL of titrant solution [500 μM $ZnCl_2$ (Nacalai Tesque, Cat# 36920-24)] were injected into protein solution (300 μL of 20 μM ZnT7) at 150 s intervals with gentle stirring at 750 rpm at 20 °C. The buffer used was 20 mM Tris–HCl, pH 7.5, 150 mM NaCl and 0.02% GDN. The temperature was kept at 20 °C throughout the measurement. The titration data were fitted to the One Set of Sites model provided by the Microcal Origin software (Origin 7 SR4 v7.0552; OriginLab Corporation). The binding enthalpy change ($\Delta H$, kcal/mol), dissociation constant ($K_d$, μM), and the binding stoichiometry (N) were permitted to float during the least-squares minimization and taken as the best-fit values.

## $Zn^{2+}$ transport assay with proteoliposomes

The $Zn^{2+}$ transport activity of hZnT7 was measured using FluoZin-3 (ThermoFisher Scientific, Cat# F24194), a $Zn^{2+}$-sensitive fluorophore. To avoid bleaching of the fluorophore, the sample was shielded from direct light throughout the experiments. Briefly, 1 mg of purified hZnT7 or hZnT7ΔHis-loop was added to 5 mg egg yolk phosphatidylcholine

(egg PC, Avanti Polar Lipids, Cat# 840051 P) in 1 mL of PBS containing 0.8% sodium cholate (Sigma-Aldrich, Cat# C6445-10G). The solution was incubated with 100 mg of wet fresh Bio-Beads SM-2 (Bio-Rad, Cat# 152-8920) overnight at 4 °C, with gentle rotation. The beads were removed, and the supernatant was centrifuged at 200,000 × g for 20 min at 4 °C. Each pellet was resuspended in 100 μL Buffer IN [20 mM HEPES (Nacalai Tesque, Cat# 17546-05), pH 6.5 and 150 mM KCl (Nacalai Tesque, Cat# 28514-75)], and mixed with 50 μM Fluozin-3. The mixtures were sonicated for 30 sec, frozen in liquid nitrogen, and thawed on ice. This procedure was repeated twice. The mixtures were again sonicated for 30 sec and applied to a PD-10 column (GE Healthcare, Cat# 17085101). The eluted fractions were centrifuged at 200,000 × g for 20 min at 4 °C. The pellets were resuspended in 100 μL Buffer IN and kept on ice prior to the assay. As a negative control, empty liposomes were prepared using the same protocol. The proteoliposomes or liposomes thus prepared were mixed with 100 μM $ZnSO_4$ (Hampton Research, Cat# HR2-245) in a cuvette containing 500 μL buffer OUT (20 mM HEPES, pH 7.5 and 150 mM KCl). Fluorescence spectra were recorded at an excitation wavelength of 490 nm using a Fluorescence Spectrophotometer F-7000 (Hitachi). To normalize the time-dependent fluorescence signal from each proteoliposome preparation (FP), 1% octyl-β-D-glucloside (OG, Dojindo, Cat# 344-05034) detergent solution in buffer OUT was mixed with the proteoliposome for 1 h on ice in the dark, and the maximal fluorescence signal ($FP_{max}$) was measured. Background signal from the protein-free liposome (FL) was also observed and normalized with the maximal fluorescence signal from the protein-free liposome treated with 1% OG detergent ($FL_{max}$). $Zn^{2+}$ transport activity was calculated as $FP/FP_{max}$ - $FL/FL_{max}$ over time, with initial transport rates ($\Delta F\,s^{-1}$) determined by linear regression of the data during the first 2 s[16]. Data analysis was performed using GraphPad Prism v9.3.1 (GraphPad Software, LLC).

### Reporting summary

Further information on research design is available in the Nature Portfolio Reporting Summary linked to this article.

## Data availability

The data that support this study are available from the corresponding authors upon request. Cryo-EM density maps of hZnT7 have been deposited in the Electorn Microscopy Data Bank (EMDB) under accession codes EMD-36048 (OF-OF), EMD-36049 (OF/Zn-OF/Zn form, i.e. $Zn^{2+}$ state 3), EMD-36050 (IF-OF form), EMD-36055 (IF/Zn-OF form, i.e. $Zn^{2+}$ state 1), EMD-36051 (IF/Zn-OF/Zn form, i.e. $Zn^{2+}$ state 2), and EMD-36053 ($Zn^{2+}$-hZnT7ΔHis-loop in OF/Zn-OF/Zn form). The atomic coordinates of human ZnT7 have been deposited in the Protein Data Bank (PDB) under accession codes 8J7T (hZnT7-Fab complex comprising two $Zn^{2+}$-unbound OF protomers; OF-OF form), 8J7U (hZnT7-Fab complex comprising two $Zn^{2+}$-bound OF protomers; OF/Zn-OF/Zn form, i.e. $Zn^{2+}$ state 3), 8J7V (hZnT7-Fab complex comprising $Zn^{2+}$-unbound IF and OF protomers; IF-OF form), 8J80 (hZnT7-Fab complex comprising $Zn^{2+}$-bound IF and $Zn^{2+}$-unbound OF protomers; IF/Zn-OF form, i.e. $Zn^{2+}$ state 1), 8J7W (hZnT7-Fab complex comprising $Zn^{2+}$-bound IF and $Zn^{2+}$-bound OF protomers; IF/Zn-OF/Zn form, i.e. $Zn^{2+}$ state 2), 8J7X ($Zn^{2+}$-unbound hZnT7ΔHis-loop-Fab complex in OF-OF form; Apo-hZnT7ΔHis-loop) and 8J7Y ($Zn^{2+}$-bound hZnT7ΔHis-loop-Fab complex in OF/Zn-OF/Zn form; $Zn^{2+}$-hZnT7ΔHis-loop). The source data underlying Fig. 7d and Supplementary Figure 1c are provided as a Source Data file. Source data are provided with this paper.

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

## Acknowledgements

We thank T. Yokoyama, K. Nanatani, J. Inoue, S. Koshiba and M. Yamamoto for management of the cryo-EM facility at Tohoku University Medical Megabank. This work was supported by funding from AMED-CREST (21gm1410006h0001) to K.I., JSPS KAKENHI to K.I. (18H03978, 21H04758 and 21H05247), Canon Medical Systems Corporation to K.K. and K.I., and the Basis for Supporting Innovative Drug Discovery and Life Science Research (BINDS) from the Japan Agency for Medical Research and Development (AMED) under grant numbers JP19am0101115 (support number: 1025), JP19am0101078 (support No. 2293), JP21am0101079 (support no. 2343) and JP22ama121038.

## Author contributions

H.B.B. performed almost all experiments, structure modeling and structure refinement. S.W. performed acquisition of cryo-EM images with CRYO-ARM300II, image processing, structure modeling and refinement and assisted in sample preparation. A.T. and M.K. acquired cryo-EM images with Talos Arctica and Titan Krios G3i. N.N., K.L., T.U. and S.I. prepared the monoclonal antibody Fab fragment recognizing hZnT7. Y.K. provided anti-PA-tag antibody beads for purification of PA-tagged hZnT7. M.I. prepared a stable expression cell line for hZnT7. H.F. and K.K. provided many insights into structures and mechanisms of SLC transporters including zinc transporters to improve the manuscript. H.B.B., S.W. and K.I. prepared the figures and wrote the manuscript. All authors discussed the results, critically read the manuscript, and approved the manuscript for submission. K.I. supervised this work.

## Competing interests

The authors declare no competing interests.
