## [Peer Review File · Nature Communications]

Cryo-EM structures of human zinc transporter ZnT7 reveal the mechanism of Zn²⁺ uptake into the Golgi apparatusReviewers' Comments:

Reviewer #1:

Remarks to the Author:

ZnT7 plays an essential role in transporting the zinc ion from cytosol into the Golgi apparatus. Here, Han et. al presents a series of cryo-EM structures of ZnT7 in the outward-facing, inward-facing and zinc bound states. Although the structures of ZnT8 and YiiP in both inward- and outward-facing conformations have been determined before, these new cryo-EM structures of ZnT7 reveal several structural features in both CTD and TMD that are distinct to ZnT8 and YiiP. Through performing mutagenesis, zinc binding and uptake experiments, the authors also proposed that the histidine-rich loop, a structural element that is unique in ZnT7, could bind zinc, thereby facilitating the zinc recruitment and transporting. Overall, the structural and functional works are interesting. By revealing the structures of ZnT7 in different conformational states, this work advances our understanding of the similarities and differences in the zinc transporting mechanism between different ZnT family proteins. All the cryo-EM structures of ZnT7 in the outward-facing conformation were resolved at relatively high quality; however, the cryo-EM structure of ZnT7 in the inward-facing conformation was determined at very poor quality. Thus, substantial revision is required before this work can be published. My specific points are:

(1) The cryo-EM density for ZnT7 in the inward-facing conformation was resolved at very poor quality. The reported overall resolution is misleading. Only the CTD of ZnT7 and bound FAB were resolved at good quality. The cryo-EM densities for the TMD of ZnT7 are rather weak and discontinuous. No clear secondary structural features in the TMD of ZnT7 could be observed. Therefore, the modeling of ZnT7 in the inward-facing conformation shown in Fig 2 is very questionable. The authors need to improve the cryo-EM map of inward-facing ZnT7 substantially. They should collect more data, perform more extensive global/local 3D classification and refinement. The author should also add a figure panel in Figure 2 to show the overall fitting of TMD into the cryo-EM map of inward-facing ZnT7 in at least two orthogonal views.

(2) Related to my last point, due to the low resolution reconstruction of ZnT7 in the inward-facing state, it is unconvincing that this is indeed the authentic inward-facing conformation of ZnT7. In addition, the authors propose that ZnT7 undergoes much smaller conformational change between inward-facing and outward-facing states, as compared to ZnT8 and YiiP. This is somewhat surprising. The identification of such small conformational change is also questionable, given that the cryo-EM map of inward-facing ZnT7 was resolved at poor quality. Furthermore, as shown in Fig. 4, it seems to me that the cytosolic tunnel for zinc transporting in the inward-facing ZnT7 is much narrower than that in ZnT8 and YiiP. Thus, it is uncertain whether such narrow tunnel could allow zinc to readily move from the cytosolic side to the zinc binding site in TMD (HDHD motif). Altogether, I think it is possible that this proposed inward-facing conformation of ZnT7 is actually not the authentic inward-facing conformation of ZnT7. Instead, this could be an instable intermediate state of ZnT7 during the structural transition from inward-facing to outward-facing. This could explain why such conformation was resolved at low resolution. Again, the authors need to improve the cryo-EM structure of inward-facing ZnT7, and provide more structural and functional evidence to convince the readers that this is indeed the authentic inward-facing conformation of ZnT7.

(3) The authors propose that the side-chains of HDHD motif undergo structural changes between zinc free and zinc bound states. Such a claim needs to be supported by the clear side-chain densities. The side-chain density shown in Fig. 3 needs to be improved for better clarity.

(4) It would be useful to prepare a supplementary figure to compare the structure of each protomer of ZnT7 and ZnT8 by overlaying them together.

(5) TM3 and TM6, two α -helices that are important for the ZnT7 dimerization, also undergo conformational change between inward-facing and outward-facing states. This is also different to ZnT8, where the TM3 and TM6 remain static between inward-facing and outward-facing states. Does

this suggest that the TMD-TMD interaction in the outward-facing conformation of ZnT7 is stronger than that in the inward-facing conformation? The authors need to show the interface between two TMDs in more detail.

Reviewer #2:

Remarks to the Author:

This manuscript addresses structural mechanisms underlying zinc transport by the Zn²⁺/H⁺ antiporter Znt7. This mammalian protein is responsible for ensuring adequate zinc supply to the Golgi apparatus for incorporation into metalloproteins. Zinc transport is thought to be driven by a pH gradient across the membrane. This manuscript presents the first structure of this protein, which joins previous structures of Znt8 and YiiP as representatives of the superfamily of Cation Diffusion Facilitators. In fact, the results include several structures of Znt7, revealing outward-facing (OF), inward-facing (IF) and zinc-bound OF states. This thorough structural characterization of Znt7 is augmented by functional assays of transport and zinc binding that address properties of a histidine-rich loop not present in the other, above-mentioned CDF transporters. Although this loop is a notable feature of a subset of CDF transporters, it has so far been poorly characterized from both structural and functional perspectives. The experimental approaches are rigorous, leading to a considerable amount of novel information and a plausible mechanism describing the transport cycle, in which the His-rich loop plays an integral role in recruiting zinc from the cytoplasm prior to transport via a classical alternating access mechanism. This article will be of wide interest to researchers generally interested in membrane transport as well as those more specifically interested in zinc biology. Although the work will be a welcome addition to the literature, there are a number of technical issues that should be addressed. In addition, I believe that the presentation would be greatly improved by general reorganization designed to highlight how these new insights relate to and extend previous results.

This work would, I believe, be easier to digest and therefore make a greater impact if the presentation was reorganized to place a greater emphasis on the relationship with previous studies of CDF transporters. Although the current manuscript eventually presents detailed comparisons to previous structures, these comparisons appear rather late in the manuscript: zinc binding in Fig. 5, pg 9 and structural comparisons in Suppl. Fig. 8, pg 13 in the penultimate section. Given familiarity that most readers will have with the literature, it seems natural to present new results in the context of past work. For example, the overall architecture of the homodimer is very much as expected with six transmembrane helices followed by a cytoplasmic domain with alpha-beta-beta-alpha-beta topology. Rather than reassure readers about this expectation, the current manuscript instead emphasizes novelties, using new terminologies to describe familiar features (e.g., S1-S6 for the membrane helices rather than the conventional TM1-TM6 and a "mushroom shape" implying fundamental differences from previous structures). The dimer interface, which has been discussed extensively in previous papers, is described early on, yet an explicit comparison is withheld until near the end of the manuscript (Suppl. Fig. 8). Segregation of membrane helices into a "scaffold" and a "helix-pair" is not so different from the architecture of YiiP and Znt8, but this comparison is never made. The assertion that previous work on Znt8 represents "non-atomic resolution" is an unnecessary and disingenuous slight, especially given the fact that current work - and basically all current cryo-EM work - in fact fails to resolve individual atoms. It is certainly worth pointing out differences in the pivot points of helices in the OF to IF conformational change, but the differences are relatively subtle and do not seem to reflect a fundamental difference in structural mechanisms. The observation that each protomer has its own transport path is also completely consistent with previous work. There are certainly novel aspects of Znt7 that will be of great interest, e.g., the His-rich loop and the dimer interface, but I believe these substantial differences will be better appreciated if the similarities are first acknowledged as a foundation for understanding specialization amongst the various CDF homologs.

The description of structures as outward facing and inward facing is asserted at an early stage without

adequately illustrating this important conformational feature: i.e., solvent accessibility of the transport site. This issue is finally dealt with in Fig. 4, but this comes 3 full pages after the initial description of the conformation. A related issue is the slight change seen in the OF conformation in the presence of zinc. The zinc-bound structure is at lower global resolution and appears to particularly suffer at the luminal end of the transmembrane domain raising a question about the confidence that one can have in this structural effect of Zn. Furthermore, the notion that this structure represents a Zn²⁺-occluded state before transition to the Zn²⁺-releasing step is not well supported by the current data. Finally, differences in the membrane helices of Znt7, Znt8 and YiiP are described as differences in their respective OF and IF states, without regard for the possibility that these structures may represent different substates, e.g., the observed difference between Zn-bound and Zn-free OF state of Znt7. It is certainly appropriate to describe these differences, but the ambiguities about the respective structures and their relationship to the transport cycle would best be acknowledged.

The reconfiguration of the transport site in the presence of zinc is interesting and in particular the 5-Å distance of His70 from the ion. This observation is surprising given the involvement of all four conserved zinc binding residues in other structures. Is it possible that a water molecule mediates an interaction between His70 and the zinc ion, thus representing an initial step in rehydration of the ion prior to its release? A related issue is the geometry of the coordination, which is not directly addressed in the manuscript. The ideal geometry would be tetrahedral and it would be interesting to know how the observed structure deviates from this idea.

The His-rich loop is not visible in the cryo-EM structures. Although this result is not surprising given the amino acid composition of the loop, this observation is not addressed until the end of page 10. It seems like it should be part of the basic description of the structures. The alpha-fold prediction is worth describing, but cannot be taken seriously in terms of 3D structure. As indicated by the sequence alignment in Fig. 1, the first four histidines align with the loop between S4 and S5 that is visible in Znt8 and YiiP, which therefore provide a structural template for the predictive algorithm. It is therefore not surprising that these residues are clustered at the cytoplasmic end of S4. The extended alpha-fold structure for the remainder of the loop, shown in Suppl. Fig. 20, indicates that alpha-fold has no idea about the actual structure. This will be evident to structural biologists, but perhaps the general reader would benefit from some comment about this unrealistic chain trace.

The penultimate paragraph describes the role of a hydrophobic gate at the cytoplasmic surface. The discussion of this gate is unconvincing as it is not clear that the two leucine residues are uniquely responsible for blocking access to the transport site (e.g., Fig 3e). Furthermore, the penultimate paragraph implies that zinc binding triggers closure of this gate to produce an occluded state, which I do not believe is supported by the current results.

Finally, there are a few technical issues with data presentation that should be addressed.

The transport assays appear to follow a standard protocol, but it would be useful to see an example of raw data to get a sense of the quality of the signal. The units used for Fig. 6 and 7 are different without explanation for this switch.

The ITC data seems to fit the narrative of the manuscript: 1 zinc site in the TM domain and 2 additional sites in the His-rich loop. However, there appears to be an endothermic signal in the ITC traces which is not mentioned and seemingly not taken into account during the analysis.

The IF structure shown in Suppl. Fig. 3 should include a surface rendering showing local resolution. Furthermore, the coloring scheme for local resolution generally map an overly broad spectrum of resolution, such that the red colors (lowest resolution) are not visible in the structure. It would be more informative to set the scale of the local resolution such that the more poorly ordered membrane domain is reddish whilst the well ordered core of the CTD/Fab is uniquely dark blue (i.e., more like Suppl. Fig. 15). Along these lines, the statement in the abstract, "We herein present the 2.8-2.9 Å-

resolution cryo-EM structures", is misleading since at least two of the structures are at 3.4 Å resolution.

Title: Cryo-EM structures of human zinc transporter ZnT7 reveal the mechanism of Zn²⁺ uptake into the Golgi apparatus

Authors: Bui B. H. et al.

Manuscript No.: NCOMMS- 22-09756A-Z

Reviewer #1:

ZnT7 plays an essential role in transporting the zinc ion from cytosol into the Golgi apparatus. Here, Han et. al presents a series of cryo-EM structures of ZnT7 in the outward-facing, inward-facing and zinc bound states. Although the structures of ZnT8 and YiiP in both inward- and outward-facing conformations have been determined before, these new cryo-EM structures of ZnT7 reveal several structural features in both CTD and TMD that are distinct to ZnT8 and YiiP. Through performing mutagenesis, zinc binding and uptake experiments, the authors also proposed that the histidine-rich loop, a structural element that is unique in ZnT7, could bind zinc, thereby facilitating the zinc recruitment and transporting. Overall, the structural and functional works are interesting. By revealing the structures of ZnT7 in different conformational states, this work advances our understanding of the similarities and differences in the zinc transporting mechanism between different ZnT family proteins.

Response: We thank the reviewer for his/her overall positive evaluation and constructive suggestions to our work. Our point-by-point responses are described below.

All the cryo-EM structures of ZnT7 in the outward-facing conformation were resolved at relatively high quality; however, the cryo-EM structure of ZnT7 in the inward-facing conformation was determined at very poor quality. Thus, substantial revision is required before this work can be published. My specific points are:

[1] The cryo-EM density for ZnT7 in the inward-facing conformation was resolved at very poor quality. The reported overall resolution is misleading. Only the CTD of ZnT7 and bound FAB were resolved at good quality. The cryo-EM densities for the TMD of ZnT7 are rather weak and discontinuous. No clear secondary structural features in the TMD of ZnT7 could be observed. Therefore, the modeling of ZnT7 in the inward-facing conformation shown in Fig 2 is very questionable. The authors need to improve the cryo-EM map of inward-facing ZnT7 substantially. They should collect more data, perform more extensive global/local 3D classification and refinement.

Response: To address this comment appropriately and convincingly, we again prepared Zn²⁺-unbound form of hZnT7 using buffer containing 1 mM EDTA throughout the purification, and acquired more than 6,000 of micrographs with the newly prepared hZnT7-Fab#1 complex sample. After thoughtful data processing using cryoSPARC and Relion3.1 in combination (for the detailed workflow of our data processing, please see new Supplementary Fig. 3), the cryo-EM map of Zn²⁺-unbound hZnT7 has reached up to 2.2 Å resolution, yielding even clearer density map for the entire region of hZnT7 as demonstrated in new Supplementary Fig. 4a-d. However, we were unable to identify a subclass of the hZnT7 homodimer with both protomers in an inward-facing (IF) conformation in the updated data analysis. Thus, this subclass may be unstable and exist as an only minor or transient population. In support of this, our predicted IF-IF homodimer model shows the much less tight or weaker interactions between the transmembrane (TM) domains at the dimer interface (new Supplementary Fig. 21), compared to those in the OF-OF homodimer (new Fig. 2c). Given the situation, we have decided to remove the original description and figure regarding the IF-IF homodimer of hZnT7 in the revised manuscript.

Of note, the newly acquired cryo-EM data enabled us to identify two types of the Zn²⁺-unbound hZnT7 dimers, a homodimer composed of two outward-facing (OF) protomers (OF-OF form) and a heterodimer composed of IF and OF protomers (IF-OF form), with high accuracy, as demonstrated in new Figs. 2 and 3, respectively. These higher-resolution cryo-EM structures of the Zn²⁺-unbound hZnT7 dimers, together with those of the Zn²⁺-bound ones, provide important mechanistic insight into hZnT7-mediated Zn²⁺-transport, as described entirely in the revised manuscript.

[2] The author should also add a figure panel in Figure 2 to show the overall fitting of TMD into the cryo-EM map of inward-facing ZnT7 in at least two orthogonal views.

Response: As aforementioned, we removed the cryo-structure of the IF-IF homodimer of hZnT7 from the revised manuscript. Therefore, we do not prepare the requested figure panel. Instead, we added new figure panels showing the overall fitting of the TMD into the cryo-EM maps of the OF-OF homodimer and IF-OF heterodimer of hZnT7 (please see new Supplementary Fig. 4a-d).

[3] Related to my last point, due to the low resolution reconstruction of ZnT7 in the inward-facing state, it is unconvincing that this is indeed the authentic inward-

inward-facing conformation of ZnT7. In addition, the authors propose that ZnT7 undergoes much smaller conformational change between inward-facing and outward-facing states, as compared to ZnT8 and YiiP. This is somewhat surprising. The identification of such small conformational change is also questionable, given that the cryo-EM map of inward-facing ZnT7 was resolved at poor quality. Furthermore, as shown in Fig. 4, it seems to me that the cytosolic tunnel for zinc transporting in the inward-facing ZnT7 is much narrower than that in ZnT8 and YiiP. Thus, it is uncertain whether such narrow tunnel could allow zinc to readily move from the cytosolic side to the zinc binding site in TMD (HDHD motif). Altogether, I think it is possible that this proposed inward-facing conformation of ZnT7 is actually not the authentic inward-facing conformation of ZnT7. Instead, this could be an instable intermediate state of ZnT7 during the structural transition from inward-facing to outward-facing. This could explain why such conformation was resolved at low resolution. Again, the authors need to improve the cryo-EM structure of inward-facing ZnT7 and provide more structural and functional evidence to convince the readers that this is indeed the authentic inward-facing conformation of ZnT7.

Response: We appreciate and accept this critical comment. We thus deleted the description about the IF-IF homodimer of hZnT7 in the revised manuscript. As described above, the newly acquired datasets allowed us to identify the IF-OF heterodimer of hZnT7 in both Zn²⁺-unbound and -bound states. We thus verified that the IF protomer of hZnT7 indeed have a wide cytosolic cavity with the negatively charged surface in contrast to the OF protomers (please see new Fig. 4b and 4c). Of more note, we found that an exceptionally long histidine-rich loop (His-loop) characteristic of hZnT7 is incorporated into the cytosolic cavity to coordinate Zn²⁺ via His¹⁶⁴, and that His¹⁶⁴ (TM4) is subsequently replaced with His²⁴⁰ (TM5), one of the His residues constituting the HDHD motif in the TM domain (please see new Fig. 4g, h, Fig. 5, and Supplementary Fig. 14). These structural findings suggest a likely role of the His-loop for Zn²⁺ recruitment to the TM Zn²⁺ binding site. Based on the new structural data well supported by the clear cryo-EM density maps (new Figs. 2-5) and systematic biophysical experiments using a set of the His-loop mutants (Fig. 6), we propose a unique mechanism of hZnT7-mediated Zn²⁺ transport in the revised manuscript.

[4] The authors propose that the side-chains of HDHD motif undergo structural changes between zinc free and zinc bound states. Such a claim needs to be supported by the clear side-chain densities. The side-chain density shown in Fig. 3 needs to be improved for better clarity.

Response: As aforementioned, our new cryo-EM datasets provided higher-resolution cryo-EM maps for the entire region of hZnT7, including the Zn²⁺-binding site in the TMD. Indeed, the side-chain density of the HDHD motif was clearly seen in the updated cryo-EM maps for both Zn²⁺-bound and -unbound states of hZnT7, as demonstrated in new Fig. 5. Notably, the distance between the C_α atoms of His⁷⁰ (TM2) and Asp²⁴⁴ (TM5) significantly increases upon the IF-to-OF form conversion, rendering the side chain of His⁷⁰ move away from Zn²⁺. As a result, the side chain of His⁷⁰ seems likely to be more mobile, resulting in the poorer density of this residue in the OF protomers (new Fig. 5c-e). Thus, we demonstrate the updated models of the Zn²⁺ coordination structure together with the improved density in new Fig. 5. Please see also our response to comment [9] by Reviewer #2.

[5] It would be useful to prepare a supplementary figure to compare the structure of each protomer of ZnT7 and ZnT8 by overlaying them together.

Response: As requested, we prepared a new figures showing the structural comparison between the hZnT7 and ZnT8 protomers (please see new Supplementary Figs. 6 and 8). Please see also our response to comments [5] and [6] by Reviewer #2.

[6] TM3 and TM6, two α -helices that are important for the ZnT7 dimerization, also undergo conformational change between inward-facing and outward-facing states. This is also different to ZnT8, where the TM3 and TM6 remain static between inward-facing and outward-facing states. Does this suggest that the TMD-TMD interaction in the outward-facing conformation of ZnT7 is stronger than that in the inward-facing conformation? The authors need to show the interface between two TMDs in more detail.

Response: We appreciate this important comment. Our updated cryo-EM structures of hZnT7 clearly demonstrate that while the OF-OF homodimer forms the tight TMD-TMD interaction between TM2 of one protomer and TM3 of another protomer (new Fig. 2c and Supplementary Fig. 7a), and that TM2 largely moves to close the luminal gate in the IF conformation, leading to the disruption of this inter-protomer interaction (new Fig. 3c and new Supplementary Fig. 7b). Notably, the IF-IF hZnT7 homodimer was not reproducibly identified in the updated cryo-EM analysis, which may suggest that the TMD-TMD interaction in the OF-OF dimer of hZnT7 is stronger than that in the IF-IF dimer, as suggested by this reviewer. In support of this, our predicted model of the hZnT7 IF-IF homodimer displays much less tight interactions at the dimer interface than the OF-OF

homodimer (new Supplementary Fig. 21), suggesting the intrinsic instability of the IF-IF homodimer conformation. The detailed TMD-TMD interactions in the newly identified OF-OF homodimer and IF-OF heterodimer are shown in new Supplementary Fig. 7a, b, and the unstable nature of the hZnT7 IF-IF homodimer has been additionally discussed in the revised main text (Discussion section, lines 375 – 379).

In this context, superimposition of the OF-OF homodimer to the IF-OF heterodimer demonstrates that, like those of ZnT8, TM3 and TM6 of hZnT7 remain almost static during the OF-to-IF conversion (please see new Fig. 3f and Supplementary Video 1). This information has also been added in the revised main text (page 7, lines 189 – 191).

Reviewer #2:

This manuscript addresses structural mechanisms underlying zinc transport by the Zn²⁺/H⁺ antiporter Znt7. This mammalian protein is responsible for ensuring adequate zinc supply to the Golgi apparatus for incorporation into metalloproteins. Zinc transport is thought to be driven by a pH gradient across the membrane. This manuscript presents the first structure of this protein, which joins previous structures of Znt8 and YiiP as representatives of the superfamily of Cation Diffusion Facilitators. In fact, the results include several structures of Znt7, revealing outward-facing (OF), inward-facing (IF) and zinc-bound OF states. This thorough structural characterization of Znt7 is augmented by functional assays of transport and zinc binding that address properties of a histidine-rich loop not present in the other, above-mentioned CDF transporters. Although this loop is a notable feature of a subset of CDF transporters, it has so far been poorly characterized from both structural and functional perspectives. The experimental approaches are rigorous, leading to a considerable amount of novel information and a plausible mechanism describing the transport cycle, in which the His-rich loop plays an integral role in recruiting zinc from the cytoplasm prior to transport via a classical alternating access mechanism. This article will be of wide interest to researchers generally interested in membrane transport as well as those more specifically interested in zinc biology. Although the work will be a welcome addition to the literature, there are a number of technical issues that should be addressed. In addition, I believe that the presentation would be greatly improved by general reorganization designed to highlight how these new insights relate to and extend previous results. This work would, I believe, be easier to digest and therefore make a greater impact if the presentation was reorganized to place a greater emphasis on the relationship with

previous studies of CDF transporters.

Response: We appreciate this overall positive evaluation and constructive comment. We totally agree that the emphasis on the relationship between the present findings and previous studies would lead to easier digestion and greater impact of this paper. For this purpose, we substantially reorganized the main text and figures in the revised manuscript, as described in our response to the next comment of this reviewer. As an additional note, we recollected several cryo-EM datasets with newly prepared samples during the revision, which yielded higher-resolution cryo-EM maps and structure models of hZnT7 in both Zn²⁺-unbound and -bound states than the original ones. Owing to these updates, the manuscript has been well supported with solid structural data and provided novel insights into mechanisms of hZnT7-mediated Zn²⁺-transport and the role of the His-loop for Zn²⁺ recruitment. Our point-by-point responses are as below:

[1] Although the current manuscript eventually presents detailed comparisons to previous structures, these comparisons appear rather late in the manuscript: zinc binding in Fig. 5, pg 9 and structural comparisons in Suppl. Fig. 8, pg 13 in the penultimate section. Given familiarity that most readers will the literature, it seems natural to present new results in the context of past work. For example, the overall architecture of the homodimer is very much as expected with six transmembrane helices followed by a cytoplasmic domain with alpha-beta-beta-alpha-beta topology. Rather than reassure readers about this expectation, the current manuscript instead emphasizes novelties, using new terminologies to describe familiar features (e.g., S1-S6 for the membrane helices rather than the conventional TM1-TM6 and a "mushroom shape" implying fundamental differences from previous structures).

Response: We agree that it would be reader-friendly and scientifically important to present the new results of the current work in the context of past works. In the revised manuscript, therefore, we begin with the amino acid sequence alignment between hZnT7, hZnT8, and bacterial YiiP (Fig.1), and subsequently present the overall structures of the Zn²⁺-unbound hZnT7 dimers (new Figs. 2 and 3) in comparison with the previously reported structures of bacterial YiiP and hZnT8 (new Supplementary Figs. 5, 6, and 8). Next, we present the cryo-EM structures of hZnT7 dimers in Zn²⁺-bound states and highlight their Zn²⁺-coordination structures (new Figs. 4 and 5). In the following sections, we compare the Zn²⁺-binding sites in the hZnT7 dimer with those in bacterial YiiP and hZnT8 dimers (new Supplementary Fig. 16). In the final section, we focus on structural and functional roles of the exceptionally long His-loop characteristic of hZnT7 and discuss the unique mechanism of hZnT7-mediated Zn²⁺ transport. Thus, we believe that

the revised manuscript properly emphasizes the structural and mechanistic similarities and differences between hZnT7, hZnT8, and bacterial YiiP, hence more effectively appeals the relationship between the current and past studies than the original manuscript.

We agree that the new terminologies (S1-S6) are not familiar to a broad range of readers, but may rather confuse them. Therefore, we renamed S1-S6 to TM1-TM6 in a conventional way in the revised main text, and showed only the helix number in all Figures with an explanatory sentence “Numbers indicate the TM helix number from the N-terminus” in the figure legends.

The term “*mushroom shape*” may sound unfamiliar as a word expressing an overall shape of membrane proteins. However, this expression seems to clearly delineate the unique structure of the hZnT7 dimer in comparison with EcYiiP and hZnT8 with a ‘V’-shaped dimeric architecture. In line with this, tight interactions are formed between both the TMDs and CTDs within the hZnT7 dimer (new Figs. 2 and 3, and new Supplementary Figs. 5), whereas the interactions between the TMDs are much less tight or not formed in the EcYiiP and hZnT8 dimers, allowing their opened V-shape structures (new Supplementary Figs. 5 and 8). To effectively appeal the different overall shape of hZnT7 from those of hZnT8 and EcYiiP, we have decided to leave the term “*mushroom shape*” as it is in the revised manuscript (page 5, lines 127 – 130).

[2] The dimer interface, which has been discussed extensively in previous papers, is described early on, yet an explicit comparison is withheld until near the end of the manuscript (Supple Fig. 8).

Response: In accordance with this suggestion, we described the different modes of interaction at the dimer interface between hZnT7, hZnT8 and EcYiiP much earlier (page 6, lines 142 – 155), with citations of new Supplementary Figs. 5, 7, and 8 in the revised main text.

[3] Segregation of membrane helices into a "scaffold" and a "helix-pair" is not so different from the architecture of YiiP and Znt8, but this comparison is never made.

Response: As the terms “scaffold domain” and “helix-pair domain” seem neither familiar nor informative to a broad range of readers, we removed these terms in the revised manuscript. Importantly, structural comparisons between hZnT7, hZnT8, and YiiP demonstrate that the TM helix rearrangement caused by the OF-to-IF conversion in hZnT7 is similar to, but not exactly the same as those in EcYiiP and hZnT8. As shown in new Supplementary Fig. 6a, the degree of inclination of TM1 and TM2 in the OF

protomer is different among hZnT7, hZnT8, and EcYiiP. Similarly, TM1 and TM5 in the IF protomer display different orientations between these three (new Supplementary Fig. 6b). As a result, the luminal cavity in the OF protomer of hZnT7 is wider than those of hZnT8 and EcYiiP, whereas the cytosolic cavity in the IF protomer of hZnT7 is narrower than those of hZnT8 and SoYiiP. These structural findings have been additionally described in the revised manuscript (page 6, lines 139-141 and pages 7-8, lines 192-205).

[4] The assertion that previous work on Znt8 represents "non-atomic resolution" is an unnecessary and disingenuous slight, especially given the fact that current work - and basically all current cryo-EM work - in fact fails to resolve individual atoms.

Response: We acknowledge that our original phrase "*Despite being at non-atomic resolution*" is an inappropriate expression, hence have omitted this phrase in the revised manuscript.

[5] It is certainly worth pointing out differences in the pivot points of helices in the OF to IF conformational change, but the differences are relatively subtle and do not seem to reflect a fundamental difference in structural mechanisms. The observation that each protomer has its own transport path is also completely consistent with previous work.

Response: Based on the updated cryo-EM structures of hZnT7, we found that hZnT7 undergoes a different pattern of the TM helix rearrangement to open the cytosolic gate in the IF conformation than hZnT8 and SoYiiP. As aforementioned, TM4 of hZnT7 shifts outward via parallel movement, whereas its TM5 moves to a lesser extent (new Fig. 3f and Supplementary Video 1). On the other hand, both TM4 and TM5 swing largely using their luminal ends as pivot points in ZnT8 and bacterial YiiP (new Supplementary Fig. 12), resulting in a wider cytosolic cavity in their IF protomers. These different structural features among the CDF superfamily members are worthwhile to note, hence have been described in the revised text (pages 7-8, lines 192 – 205). We agree that the observation for the presence of a transport path in each protomer is consistent with previous works, and have added this information with citations of the related previous papers (page 5, lines 130-133).

[6] The description of structures as outward facing and inward facing is asserted at an early stage without adequately illustrating this important conformational feature: i.e., solvent accessibility of the transport site. This issue is finally dealt with in Fig. 4,

but this comes 3 full pages after the initial description of the conformation.

Response: In response to this comment, we have prepared new Fig. 3f and supplementary video 1 that clearly illustrate the TM helix rearrangements during the OF-to-IF form conversion and discussed this issue earlier in the revised text (page 7, lines 182-191). Simultaneously, we discuss the different solvent accessibilities of the transport site between the IF and OF protomers using new Fig. 3d and e.

[7] A related issue is the slight change seen in the OF conformation in the presence of zinc. The zinc-bound structure is at lower global resolution and appears to particularly suffer at the luminal end of the transmembrane domain raising a question about the confidence that one can have in this structural effect of Zn. Furthermore, the notion that this structure represents a Zn²⁺-occluded state before transition to the Zn²⁺-releasing step is not well supported by the current data.

Response: As shown in new Supplementary Fig. 10c, d, the updated cryo-EM structures verified that the OF protomer maintains almost the same TM helix arrangement before and after Zn²⁺ binding, allowing us to conclude that Zn²⁺ does not largely alter the conformation of the TMD in the OF protomer. This information has additionally been described in the revised main text (page 9, lines 232-235). Regarding the transient formation of the occluded state, we acknowledge that the present cryo-EM analysis reveals the static structures of hZnT7 in the IF and OF conformations, and does not capture the transiently formed “occluded state” with both cytosolic and luminal cavities closed. Thus, we have removed the description about the occluded state of hZnT7 in the revised manuscript.

[8] Finally, differences in the membrane helices of Znt7, Znt8 and YiiP are described as differences in their respective OF and IF states, without regard for the possibility that these structures may represent different substates, e.g., the observed difference between Zn-bound and Zn-free OF state of Znt7. It is certainly appropriate to describe these differences, but the ambiguities about the respective structures and their relationship to the transport cycle would best be acknowledged.

Response: We thank the reviewer for this essential comment. As described in the revised manuscript, our updated cryo-EM analysis identified two sorts of substate structures for the Zn²⁺-unbound hZnT7 dimer and three sorts of substate structures for Zn²⁺-bound hZnT7 dimer (new Figs. 2, 3, and 4). Structural comparisons of these hZnT7 substates

with the corresponding states of hZnT8 and YiiP have been made, which clarified unique structural features of the TM domain and Zn²⁺-binding site of hZnT7, as described in the revised text (page 6, lines 139-168; page 7-8, lines 192-205; page 10, lines 268-285) with citations of new Supplementary Figs. 5-8, and 16.

Regarding the relationship of the respective structures of hZnT7 to the transport cycle, we discuss a likely mechanism of Zn²⁺ uptake mediated by the hZnT7 protomer in the revised text (page 15, lines 414-428). The updated cryo-EM structures seem to nicely delineate the conformational changes during the Zn²⁺-recruitment and subsequent Zn²⁺-transport within the protomer. However, how the two protomers interplay with each other along the transport cycle remain an important question. To elucidate this essential issue, other structural approaches including the time-resolved cryo-EM analysis are necessary. Such limitation of the present study has been additionally discussed in the last paragraph of Discussion section (page 15, lines 429-435).

[9] The reconfiguration of the transport site in the presence of zinc is interesting and in particular the 5-A distance of His70 from the ion. This observation is surprising given the involvement of all four conserved zinc binding residues in other structures. Is it possible that a water molecule mediates an interaction between His70 and the zinc ion, thus representing an initial step in rehydration of the ion prior to its release? A related issue is the geometry of the coordination, which is not directly addressed in the manuscript. The ideal geometry would be tetrahedral and it would be interesting to know how the observed structure deviates from this idea.

Response: The updated cryo-EM analysis of Zn²⁺-bound hZnT7 (new Fig. 5a-d) substantially improved the density around the TM Zn²⁺-binding site, indicating that a tetrahedral Zn²⁺ complex is formed by His⁷⁰, Asp⁷⁴, Asp²⁴⁴, and His¹⁶⁴ (in Zn²⁺ state 1)/His²⁴⁰ (in Zn²⁺ state 2) in the IF protomer (new Fig. 5a, b). Interestingly, the updated cryo-EM maps demonstrate that the side chain of His⁷⁰ moves away from Zn²⁺ in the OF conformation, concomitant with significant increase of the distance between the C α atoms of His⁷⁰ and Asp²⁴⁴. Considering that the N δ /N ϵ atom of His⁷⁰ is located beyond the range of direct Zn²⁺ coordination and that some extra density is seen between these two in the Zn²⁺ state 3 (new Fig. 5c), a water molecule seems likely to exist to mediate the interaction between His⁷⁰ and Zn²⁺ in this Zn²⁺-bound OF protomer. These new structural insights and the possible role of the intermediary water molecule prior to Zn²⁺ release have been additionally discussed in the main text (page 10, lines 261-267). Please see also our response to comment [4] by reviewer #1.

[10] The His-rich loop is not visible in the cryo-EM structures. Although this result is not surprising given the amino acid composition of the loop, this observation is not addressed until the end of page 10.

Response: In response to this comment, we added a brief explanation about the location and visible density of the His-loop in the first Result section (page 6, lines 150 – 154) with a citation of new Fig. 2a,b, e.

[11] It seems like it should be part of the basic description of the structures. The alpha-fold prediction is worth describing, but cannot be taken seriously in terms of 3D structure. As indicated by the sequence alignment in Fig. 1, the first four histidines align with the loop between S4 and S5 that is visible in Znt8 and YiiP, which therefore provide a structural template for the predictive algorithm. It is therefore not surprising that these residues are clustered at the cytoplasmic end of S4. The extended alpha-fold structure for the remainder of the loop, shown in Suppl. Fig. 20, indicates that alpha-fold has no idea about the actual structure. This will be evident to structural biologists, but perhaps the general reader would benefit from some comment about this unrealistic chain trace.

Response: As this reviewer well understands, our purpose of showing the AlphaFold2 structure of hZnT7 was just to display a possible location of the His loop and the wide distribution of His residues within the loop. We agree with the reviewer that AlphaFold2 does not precisely reflect the actual structure of the His-loop in hZnT7 and that the predicted structure cannot be taken seriously. After careful consideration, we have decided to remove the AlphaFold2 structure of hZnT7 (original Supplementary Fig. 20) and its description in the revised manuscript.

[12] The penultimate paragraph describes the role of a hydrophobic gate at the cytoplasmic surface. The discussion of this gate is unconvincing as it is not clear that the two leucine residues are uniquely responsible for blocking access to the transport site (e.g., Fig 3e). Furthermore, the penultimate paragraph implies that zinc binding triggers closure of this gate to produce an occluded state, which I do not believe is supported by the current results.

Response: The amino acid sequence alignment (Fig. 1) shows that Leu²³⁹ (TM5) and Leu²⁸⁵ (TM6) of hZnT7 align well with Leu¹⁵² and Met¹⁹⁷ in EcYiiP and Leu¹⁵⁴ and Leu¹⁹⁹ in SoYiiP, residues that are known to constitute the hydrophobic gate in these bacterial

Zn²⁺ transporters. The corresponding residues in hZnT8 are Val (TM5) and Ile (TM6) (Fig. 1), suggesting that a similar hydrophobic gate may be formed near the cytosolic end of the metal transport pathway in the mammalian ZnTs. However, given that further explorations such as mutation studies have not been made regarding the roles of Leu²³⁹ and Leu²⁸⁵ in this study and that the density of Leu²³⁹ is unclear in the cryo-EM maps of both the IF and OF protomers of hZnT7, we acknowledge that the discussion about the possible role of the Leu²³⁹-Leu²⁸⁵ pair as the hydrophobic gate is still premature. We hence decided to remove it in the revised manuscript.

[13] Finally, there are a few technical issues with data presentation that should be addressed. The transport assays appear to follow a standard protocol, but it would be useful to see an example of raw data to get a sense of the quality of the signal. The units used for Fig. 6 and 7 are different without explanation for this switch.

Response: As requested, examples of raw data of the Zn²⁺ transport assay using the hZnT7 proteoliposomes were additionally shown in new supplementary Fig 20. The unit of the Zn²⁺ uptake rate has been unified to '(1/s) per µg of protein' in new Fig. 6d, where the Zn²⁺ uptake rates of hZnT7 WT, HS1, HS2, HS3, and ΔHis-loop mutants are all compiled.

[14] The ITC data seems to fit the narrative of the manuscript: 1 zinc site in the TM domain and 2 additional sites in the His-rich loop. However, there appears to be an endothermic signal in the ITC traces which is not mentioned and seemingly not taken into account during the analysis.

Response: We thank the reviewer for carefully checking our ITC data for Zn²⁺ titration to hZnT7 WT and a series of its mutants. In response to this comment, we analyzed the details of the ITC signals the reviewer pointed out. The enlarged views of these peaks (please see the next page) show that after each injection, a sharp endothermic spike was observed, followed by the slow exothermic changes. Although the small endothermic spikes appear after Zn²⁺ binding to hZnT7 reaches the saturation level, the spike reaction seem to actually occur at every injection. In general, the small sharp spike (either endothermic or exothermic) immediately after injection is thought to be derived from 1) buffer mismatch between the sample and the syringe or 2) bubble contamination. The slow heat changes likely represent heat derived from the ligand dilution. In some other cases, exothermic spikes are first observed, and then small endothermic changes are observed. These reactions are often observed in ITC measurements, and less noticeable

than major reactions (i.e. protein-ligand binding) in most cases. Therefore, these small heat changes are usually treated as background noises or experimental noises of measurements. The observed small difference between hZnT7 WT and its mutants are likely due to small difference in prepared buffers, not due to their physicochemical or functional properties.

To validate our above interpretation, we performed a control ITC experiment by titrating 0.5 mM $ZnCl_2$ to the ITC buffer (20 mM Tris-HCl pH 7.5, 150 mM NaCl, 0.02%

GDN). More concretely, 2.5 ml of the ITC buffer was concentrated using a 50 kDa filter to around 50 – 80 μ l. The concentrated buffer was then diluted with the ITC buffer to a final volume of 300 μ l, as is the same with the hZnT7 sample preparation. The buffer was titrated with 0.5 mM ZnCl₂ (diluted in the ITC buffer) at 20°C by 40 times injection. Similarly, 300 μ l of the ITC buffer (never concentrated) was also titrated. As shown below, we thus observed the enhanced endothermic signals when using the concentrated buffer (right panel) instead of the non-concentrated ITC buffer (left panel).

As an additional note, to further verify that two molar equivalents of Zn²⁺ binds to the His-loop of hZnT7, we additionally prepared a hZnT7 AAAA mutant in which the HDHD motif in the TM domain were all mutated to Ala (A), and performed the ITC measurement with it. As expected, the result indicated that this mutant bound nearly 2 molar equivalents of Zn²⁺ most likely via the His-loop. This supportive data has been added to new Fig. 6a, b and described in the revised manuscript (page 12, lines 325-328).

[15] The IF structure shown in Suppl. Fig. 3 should include a surface rendering showing local resolution. Furthermore, the coloring scheme for local resolution generally map an overly broad spectrum of resolution, such that the red colors (lowest resolution) are not visible in the structure. It would be more informative to set the scale of the local resolution such that the more poorly ordered membrane domain is reddish whilst the well ordered core of the CTD/Fab is uniquely dark blue

(i.e., more like Suppl. Fig. 15). Along these lines, the statement in the abstract, "We herein present the 2.8-2.9 Å-resolution cryo-EM structures", is misleading since at least two of the structures are at 3.4 Å resolution.

Response: We again appreciate this useful comment. Based on the updated cryo-EM data, we recalculated the local resolution maps of the hZnT7-Fab complexes and displayed them with the appropriate scale of the local resolution in new Supplementary Figs. 3, 13, 15, and 18. In the improved maps, the core regions of hZnT7 and Fab are shown in dark blue or light green while the poorly ordered TM regions are shown in red. We also carefully revised statements about the resolutions of the current cryo-EM maps throughout the text.

Reviewers' Comments:

Reviewer #1:

Remarks to the Author:

The authors have performed substantial work during the revision. They have collected new cryo-EM data and revealed a heterodimer of ZnT7 with one subunit adopting outward-facing conformation and another adopting inward-facing conformation. Similar heterogeneous conformation was also observed in the structure of ZnT8. This needs to be mentioned in this paper. Overall, the cryo-EM work has been improved. I therefore support the publication of this work at Nature Communications.

Reviewer #2:

Remarks to the Author:

This is a much improved manuscript describing structures of Znt7 in combination with biophysical characterization of Zn binding and transport. For the revision, the authors have included higher resolution structures of a wider variety of conformational states with Zn present and absent at the transport site. Altogether, it is a comprehensive description of the molecule with reasonable reference to preceding work on other members of the cation diffusion facilitator family. I have only a few minor suggestions for improvement.

Line 129 and Suppl Fig. 5 compare the membrane interface between dimers of Znt7, Znt8 and EcYiiP, the latter of which have looser interactions. Why not include soYiiP which displays a tight interaction like with Znt7?

Line 154 implies that model building included the His-rich loop, but line 152 states that it was invisible. This apparent contradiction should be clarified. How much of the His-rich loop was actually modeled (e.g., number of residues)?

The word "trigonometry" on line 280 is incorrect. I believe the authors mean geometry.

In the first paragraph of the Discussion the authors discuss differences seen at 10 μ M vs. 200-300 μ M Zn. This might be a good place to acknowledge that the concentration of "free" Zn in the cytosol is well below 10 μ M, estimated by Outten & O'Halloran (2001) to be in the picomolar range. Discrepancy with measured K_d and K_m values of transport proteins remains a central enigma in Zn biology and in my opinion should be highlighted as a major ambiguity in our understanding of how Zn transporters function in a physiological environment.

Fig 7 is effective in showing the structural features revealed in this study. However, readers might appreciate depiction of other features relevant to the transport cycle, such as the negative electrostatic charge of the binding site in the IF conformation, protonation/hydration of His70 in the OF conformation. As depicted, there is no clear sense of Zn export nor of the antiport mechanism.

Title: Cryo-EM structures of human zinc transporter ZnT7 reveal the mechanism of Zn²⁺ uptake into the Golgi apparatus

Authors: Bui, B. H. et al.

Manuscript No.: NCOMMS- 22-09756A-Z

Reviewer #1 (Remarks to the Author):

The authors have performed substantial work during the revision. They have collected new cryo-EM data and revealed a heterodimer of ZnT7 with one subunit adopting outward-facing conformation and another adopting inward-facing conformation. Similar heterogeneous conformation was also observed in the structure of ZnT8. This needs to be mentioned in this paper. Overall, the cryo-EM work has been improved. I therefore support the publication of this work at Nature Communications.

Response: We thank the reviewer for his/her positive evaluation and additional constructive comment to our work. As requested, we have added information on the heterogeneous conformation observed in the structure of hZnT8 to the revised text (lines 178-179, page 7).

Reviewer #2 (Remarks to the Author):

This is a much improved manuscript describing structures of Znt7 in combination with biophysical characterization of Zn binding and transport. For the revision, the authors have included higher resolution structures of a wider variety of conformational states with Zn present and absent at the transport site. Altogether, it is a comprehensive description of the molecule with reasonable reference to preceding work on other members of the cation diffusion facilitator family. I have only a few minor suggestions for improvement.

Response: We thank the reviewer for his/her positive evaluation and additional constructive comments to our work. Our point-by-point responses are described below.

[1] Line 129 and Suppl Fig. 5 compare the membrane interface between dimers of Znt7, Znt8 and EcYiiP, the latter of which have looser interactions. Why not include

soYiiP which displays a tight interaction like with Znt7?

Response: As suggested, we have added the cryo-EM structure of the SoYiiP dimer (PDB ID: 7KZZ) in the Supplementary Fig. 5, and mentioned about its tight inter-protomer interactions in the revised text (lines 127-128, page 5).

[2] Line 154 implies that model building included the His-rich loop, but line 152 states that it was invisible. This apparent contradiction should be clarified. How much of the His-rich loop was actually modeled (e.g., number of residues)?

Response: As requested, we have added information on the His-rich loop segments visible in the cryo-EM map (e.g., residues 159-163 and 229-232), in the revised manuscript (lines 151-154, page 6 and lines 292-293, pages 10 and 11).

[3] The word "trigonometry" on line 280 is incorrect. I believe the authors mean geometry.

Response: The original sentence including the word "trigonometry" has been corrected in the revised manuscript (lines 281-282, page 10) as follows:

“whereas bacterial YiiP forms a binuclear zinc coordination with the (HHD)₂ motif at the CTD dimer interface”

In the previous paper [Lu et al. Nat. Struct. Mol. Biol. 16, 1063-1067 (2009)], indeed, the word “a binuclear zinc coordination” was used to express the Zn²⁺-binding site at the CTD dimer interface of *E. coli* YiiP.

[4] In the first paragraph of the Discussion the authors discuss differences seen at 10 uM vs. 200-300 uM Zn. This might be a good place to acknowledge that the concentration of "free" Zn in the cytosol is well below 10 uM, estimated by Outten & O'Halloran (2001) to be in the picomolar range. Discrepancy with measured Kd and Km values of transport proteins remains a central enigma in Zn biology and in my opinion should be highlighted as a major ambiguity in our understanding of how Zn transporters function in a physiological environment.

Response: We thank the reviewer for this critical comment. As suggested, we, in the revised manuscript, additionally discussed the discrepancy between our determined K_a for Zn²⁺ value of hZnT7 and the physiological labile Zn²⁺ concentration as a central

enigma in mechanisms of zinc transporters. Thus, we inserted the following sentences into the first paragraph of Discussion (lines 372-379, page 13):

“Indeed, our ITC analysis showed that the K_d for Zn^{2+} value of purified hZnT7 is around 10 μM (Fig. 7). However, Outten & O'Halloran (2001) estimated the labile Zn^{2+} concentration in the cytosol to be in the picomolar range, much below 10 μM ²⁰. More recently, Liu et al. (2022) reported that the cytosolic labile Zn^{2+} concentration in HeLa cells is ~ 0.13 nM²¹. Thus, we need to carefully consider the discrepancy between the Zn^{2+} -binding affinity of hZnT7 determined by the in vitro biophysical approach and the Zn^{2+} availability in the physiological environment as a central enigma in mechanisms of zinc transporters.”

[5] Fig 7 is effective in showing the structural features revealed in this study. However, readers might appreciate depiction of other features relevant to the transport cycle, such as the negative electrostatic charge of the binding site in the IF conformation, protonation/hydration of His70 in the OF conformation. As depicted, there is no clear sense of Zn export nor of the antiport mechanism.

Response: We appreciate this constructive comment. As suggested, we modified Fig. 8 (Fig. 7 originally) so that this figure and its legend contain additional information on the negatively charged surface of the cytosolic Zn^{2+} entry cavity in the IF conformation and the likely protonation of His70 in the OF conformation.